# CTRLORA: AN EXTENSIBLE AND EFFICIENT FRAMEWORK FOR CONTROLLABLE IMAGE GENERATION

**Yifeng Xu**[1,2], **Zhenliang He**[1], **Shiguang Shan**[1,2], **Xilin Chen**[1,2]

[1]Key Lab of AI Safety, Institute of Computing Technology, CAS, China
[2]University of Chinese Academy of Sciences, China
`yifeng.xu@vipl.ict.ac.cn`, `{hezhenliang,sgshan,xlchen}@ict.ac.cn`

## ABSTRACT

Recently, large-scale diffusion models have made impressive progress in text-to-image (T2I) generation. To further equip these T2I models with fine-grained spatial control, approaches like ControlNet introduce an extra network that learns to follow a condition image. However, for every single condition type, ControlNet requires independent training on millions of data pairs with hundreds of GPU hours, which is quite expensive and makes it challenging for ordinary users to explore and develop new types of conditions. To address this problem, we propose the CtrLoRA framework, which trains a *Base ControlNet* to learn the common knowledge of image-to-image generation from multiple base conditions, along with *condition-specific LoRAs* to capture distinct characteristics of each condition. Utilizing our pretrained Base ControlNet, users can easily adapt it to new conditions, requiring as few as 1,000 data pairs and less than one hour of single-GPU training to obtain satisfactory results in most scenarios. Moreover, our CtrLoRA reduces the learnable parameters by 90% compared to ControlNet, significantly lowering the threshold to distribute and deploy the model weights. Extensive experiments on various types of conditions demonstrate the efficiency and effectiveness of our method. Codes and model weights will be released at `https://github.com/xyfJASON/ctrlora`.

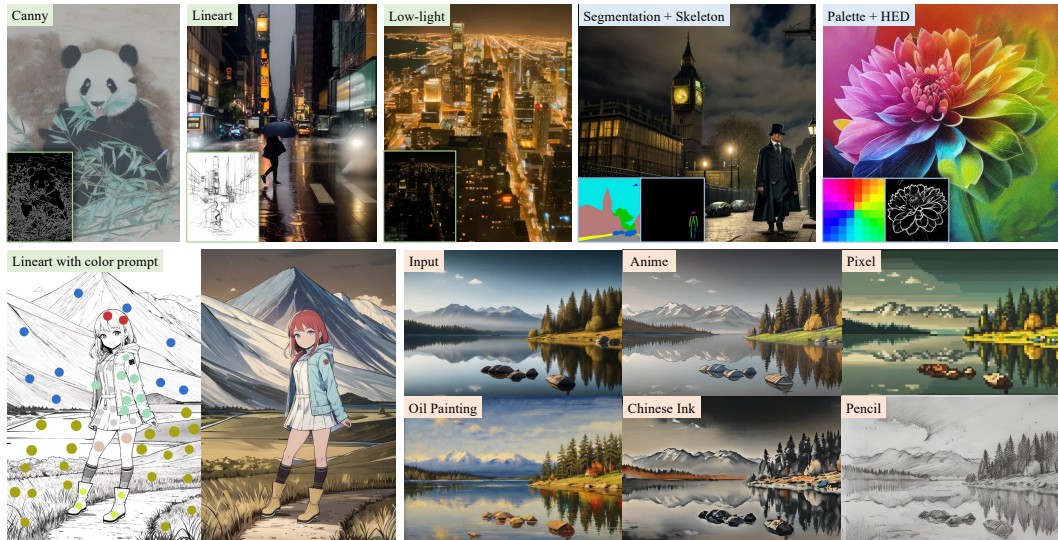

Figure 1: Our results of single-conditional generation, multi-conditional generation, style transfer.

## 1 INTRODUCTION

In recent years, diffusion models (Sohl-Dickstein et al., 2015; Song & Ermon, 2019; Ho et al., 2020) have become one of the most popular generative models for visual generation and editing tasks. The superior performance and scalability of diffusion models encourage researchers to train large ones on billions of text-image pairs (Schuhmann et al., 2022), resulting in powerful and influential text-to-image (T2I) *base models* (Rombach et al., 2022; Saharia et al., 2022; Ramesh et al.,

2022; Betker et al., 2023; Chen et al., 2024; Xue et al., 2024). Further, by combining these base models with *parameter-efficient fine-tuning (PEFT)* methods such as LoRA (Hu et al., 2022; Ryu, 2022), users can obtain personalized models without the need for a large amount of training data and computational resources, which significantly lowers the threshold for extending a T2I base model to the creation of various kinds of art. Leveraging the "Base + PEFT" paradigm, especially "Stable Diffusion + LoRA", numerous individuals from both technical and non-technical backgrounds including artists, have embraced this methodology for artistic creation, forming a huge community and technological ecosystem.

However, it is challenging for T2I models to accurately control spatial details such as layout and pose, as text prompts alone are not precise enough to convey these specifics. To solve this problem, ControlNet (Zhang et al., 2023) adds an extra network that accepts a condition image, turning a T2I model into an image-to-image (I2I) model. In this manner, ControlNet is able to generate images according to a specific kind of condition image such as canny edge, significantly improving the controllability. However, for each condition type, an independent ControlNet needs to be trained from scratch with a large amount of data and computational resources. For example, the ControlNet conditioned on canny edge is trained on 3 million images for around 600 A100 GPU hours. This substantial budget makes it challenging for ordinary users to create a ControlNet for a novel kind of condition image, hindering the growth of the ControlNet community compared to the flourishing Stable Diffusion community[1]. Therefore, it is worth figuring out a simple and economical solution to extend the promising ControlNet to handle novel kinds of condition images.

To address this problem, inspired by the "Base + PEFT" paradigm, we propose a CtrLoRA framework that allows users to conveniently and efficiently establish a ControlNet for a customized type of condition image. As illustrated in Fig. 2(a), we first train a *Base ControlNet* on a large-scale dataset across multiple base condition-to-image tasks such as canny-to-image, depth-to-image, and skeleton-to-image, where the network parameters are shared by all these base conditions. Meanwhile, for each base condition, we add a *condition-specific LoRA* to the Base ControlNet. In this manner, the condition-specific LoRAs capture the unique characteristics of the corresponding conditions, allowing the Base ControlNet to focus on learning the common knowledge of image-to-image (I2I) generation from multiple conditions simultaneously. Therefore, a well-trained Base ControlNet with general I2I ability can be easily extended to any novel condition by training new LoRA layers, as shown in Fig. 2(b). With our framework, in most scenarios, we can learn a customized type of condition with as few as 1,000 training data and less than one hour of training on a single GPU. Moreover, our method requires only 37 million LoRA parameters per new condition, a significant reduction compared to the 361 million parameters required by the original ControlNet for each condition. In a word, our method substantially lowers the resource requirements compared to the original ControlNet, as detailed in Table 1.

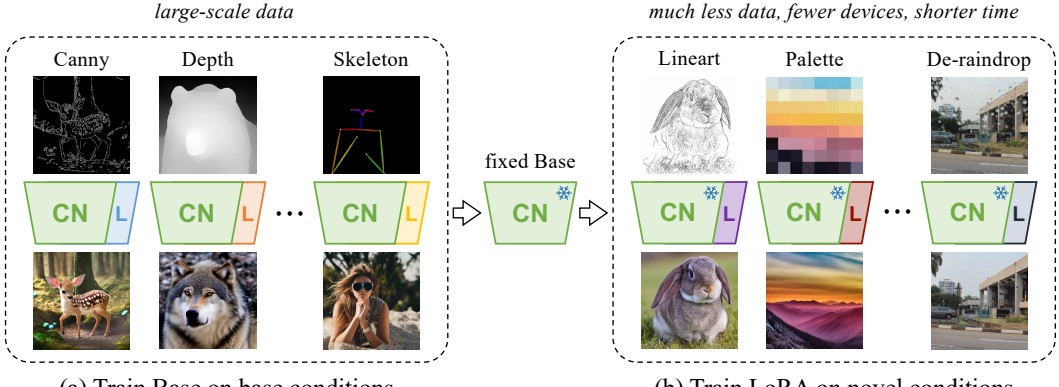

(a) Train Base on base conditions          (b) Train LoRA on novel conditions

Figure 2: Overview of the CtrLoRA framework. "CN" denotes Base ControlNet, "L" denotes LoRA. (a) We first train a shared Base ControlNet in conjunction with condition-specific LoRAs on a large-scale dataset that contains multiple base conditions. (b) The trained Base ControlNet can be easily adapted to novel conditions with significantly less data, fewer devices, and shorter time.

---

[1] As of September 24, 2024, on civitai.com, one of the most popular repositories for AI art models, there are 1024 models tagged with Stable Diffusion whereas only 56 are tagged with ControlNet.

Table 1: Comparison of model size, dataset size, and training time cost. For $N$ conditions, the total number of parameters is $361M \times N$ for ControlNet and $360M + 37M \times N$ for our CtrLoRA.

| Method | Condition | # Params. | Dataset Size | GPU Hours |
|---|---|---|---|---|
| ControlNet | Canny | 361M | 3M (Internet) | $\sim 600$ (A100) |
| | Depth | 361M | 3M (Internet) | $\sim 500$ (A100) |
| | Skeleton | 361M | 200K (Openpose) | $\sim 300$ (A100) |
| | ... | ... | ... | ... |
| CtrLoRA (Base) | 9 base conditions | 360M | 20M (MultiGen) | $\sim 6000$ (4090) |
| CtrLoRA (LoRA) for novel conditions | Lineart | 37M | 1K (Custom) | $\sim 0.17$ (4090) |
| | Palette | 37M | 1K (Custom) | $\sim 0.83$ (4090) |
| | De-raindrop | 37M | 863 (Raindrop) | $\sim 0.83$ (4090) |
| | ... | ... | ... | ... |

Our contributions are summarized below:

1. We propose an effective and efficient framework for extensible image-to-image generation. This framework utilizes a shared Base ControlNet to learn the common knowledge of image-to-image generation, while employing condition-specific LoRAs to capture unique characteristics of each image-to-image task.

2. Our Base ControlNet can be easily and economically adapted to novel conditions by training new LoRA layers, which requires significantly fewer resources compared to the original ControlNet, including reduced training data, shortened training time, and decreased model size. As a result, our method considerably lowers the barrier for ordinary users to create a customized ControlNet.

3. Without extra training, our Base ControlNet and LoRAs can be seamlessly integrated into various Stable Diffusion based models from the public community. Moreover, the LoRAs trained for different conditions can be easily combined for finer and more complex control.

4. We optimize the design and initialization strategy of the condition embedding network, which significantly accelerates the training convergence. Furthermore, in this way, we do not observe the phenomenon of sudden convergence that appears in the original ControlNet.

## 2 RELATED WORK

**Diffusion models.** Diffusion models, originally introduced by Sohl-Dickstein et al. (2015) and substantially developed by Song & Ermon (2019); Ho et al. (2020); Song et al. (2021b); Dhariwal & Nichol (2021); Ho & Salimans (2022); Bao et al. (2023); Peebles & Xie (2023), etc., have gained widespread popularity as a type of generative model. To further enhance the expressiveness of diffusion models, researchers proposed to model the diffusion process in the latent space (Vahdat et al., 2021; Rombach et al., 2022) of a variational autoencoder (Kingma & Welling, 2013), which enables high-resolution image generation. The proposed CtrLoRA in this paper is built upon Stable Diffusion (Rombach et al., 2022), a widely used latent diffusion model.

**Conditional generation.** To advance text-to-image (T2I) generation, researchers incorporate the text embedding from CLIP (Radford et al., 2021) or T5 (Raffel et al., 2020) into diffusion models, leading to powerful large-scale T2I models (Nichol et al., 2022; Ramesh et al., 2022; Rombach et al., 2022; Balaji et al., 2022; Saharia et al., 2022). To facilitate more fine-grained control, several methods (Li et al., 2023; Zhang et al., 2023) inject spatial conditions into the model, significantly enhancing the controllability. For example, ControlNet (Zhang et al., 2023) introduces an auxiliary network to process the condition images and integrates this network into the Stable Diffusion model. However, training a ControlNet for each single condition requires large amounts of data and time, creating a considerable burden. To address this problem, T2I-Adapter (Mou et al., 2024), SCEdit (Jiang et al., 2024), and ControlNet-XS (Zavadski et al., 2023) design efficient network architectures to reduce the model size, but training these models still requires large-scale datasets and devices. ControlLoRA (Hecong, 2023) directly trains LoRAs with conditional input on Stable Diffusion, but suffers from suboptimal performance with limited data. X-Adapter (Ran et al., 2024)

and Ctrl-Adapter (Lin et al., 2024) leverage pretrained ControlNets and efficiently adapt them to up-graded backbones. UniControl (Qin et al., 2024) and Uni-ControlNet (Zhao et al., 2024) train a unified model to manage multiple conditions, significantly reducing the number of models. However, these two methods lack a straightforward and convenient manner for users to add new conditions, which limits their practicality in real-world scenarios. In contrast, our method can efficiently learn new conditions with significantly less data and fewer resources.

**Low-Rank Adaptation.** Low-Rank Adaptation (LoRA) is a well-known technique for parameter-efficient fine-tuning of large language models (Hu et al., 2022) and image generation models (Ryu, 2022). This method follows the assumption that the updates to the model weights have a low "intrinsic rank" during fine-tuning, which can be represented by a low-rank decomposition $\Delta W = BA$, where $B \in \mathbb{R}^{d \times r}, A \in \mathbb{R}^{r \times d}$, and $r \ll d$. In practice, LoRA significantly reduces the number of optimizable parameters while maintaining a decent performance. Although LoRA is widely used in conjunction with Stable Diffusion (Rombach et al., 2022) for customized image generation, it is rarely used with another prominent image generation technique, ControlNet (Zhang et al., 2023). We think the main reason is that the ControlNet is trained separately for different conditions, making it unsuitable to serve as a foundation model that can be shared across various conditions. In this paper, we investigate a novel method to train a Base ControlNet as a foundation model in cooperation with the LoRA technique.

## 3 METHOD

In this section, we introduce the design and training strategy of our CtrLoRA framework for extensible image-to-image (I2I) generation. In Section 3.1, we present the fundamental formulations and clarify the associated notations. In Section 3.2, we propose Base ControlNet that serves as a foundation model for various I2I generation tasks. In Section 3.3, we illustrate how to efficiently adapt our Base ControlNet to new conditions with LoRAs. In Section 3.4, we explain our design of the condition embedding network to accelerate the training convergence.

### 3.1 PRELIMINARIES

In diffusion models (Ho et al., 2020; Rombach et al., 2022), each data sample $\mathbf{x}_0$ is diffused into Gaussian noise through a Markov process, while a generative model is trained to reverse this process with the following loss function:

$$\mathcal{L}(\theta) = \mathbb{E}_{\mathbf{x}_0 \sim p_{data}, t \sim U(0,T), \boldsymbol{\epsilon} \sim \mathcal{N}(\mathbf{0}, \mathbf{I})} \left[ \left\| \boldsymbol{\epsilon} - \boldsymbol{\epsilon}_\theta(\mathbf{x}_t) \right\|^2 \right] \tag{1}$$

where $\mathbf{x}_t = \sqrt{\bar{\alpha}_t}\mathbf{x}_0 + \sqrt{1 - \bar{\alpha}_t}\boldsymbol{\epsilon}$ denotes the noised sample at step $t$, and $\boldsymbol{\epsilon}_\theta$ is a neural network designed to predict the diffusion noise $\boldsymbol{\epsilon}$.

For conditional generation, the loss function can be modified as follows (Ho & Salimans, 2022):

$$\mathcal{L}(\theta) = \mathbb{E}_{(\mathbf{x}_0, \mathbf{c}) \sim p_{data}, t \sim U(0,T), \boldsymbol{\epsilon} \sim \mathcal{N}(\mathbf{0}, \mathbf{I})} \left[ \left\| \boldsymbol{\epsilon} - \boldsymbol{\epsilon}_\theta(\mathbf{x}_t, \mathbf{c}) \right\|^2 \right] \tag{2}$$

where $\mathbf{c}$ denotes the conditional signal such as text for text-to-image generation and image for image-to-image (I2I) generation[2]. Specifically, ControlNet (Zhang et al., 2023) for I2I generation designs the noise prediction network $\boldsymbol{\epsilon}_\theta(\mathbf{x}_t, \mathbf{c})$ as follows:

$$\boldsymbol{\epsilon}_\theta(\mathbf{x}_t, \mathbf{c}) = \mathcal{D}\left( \mathcal{E}(\mathbf{x}_t), \mathcal{C}_\theta(\mathbf{x}_t, \mathcal{F}_\theta(\mathbf{c})) \right) \tag{3}$$

where $\mathcal{E}$ and $\mathcal{D}$ denote the encoder and decoder of the UNet pretrained in Stable Diffusion (Rombach et al., 2022), $\mathcal{C}_\theta$ denotes the ControlNet, and $\mathcal{F}_\theta$ denotes the condition embedding network.

In the original ControlNet, $\mathcal{C}_\theta$ in Eq. (3) is independently trained for each type of condition image and cannot be shared across different conditions, which leads to a huge demand for training data and computational resources. In the following sections, we introduce our CtrLoRA framework that trains $\mathcal{C}_\theta$ as a shared and extensible Base ControlNet, and explain how to efficiently extend it to various new conditions with much less data and fewer devices.

---

[2]In the following, we focus on I2I generation, and the text condition is assumed to be the default. Therefore, to simplify the notation, we omit the text condition and use $\mathbf{c}$ to represent the image condition.

## 3.2 BASE CONTROLNET FOR EXTENSIBLE I2I GENERATION

A generalizable model for various image-to-image (I2I) generation tasks necessitates a comprehensive understanding of I2I generation. To this end, we propose to train a shared *Base ControlNet* across multiple types of condition images simultaneously, in order to acquire common knowledge of diverse I2I tasks. Meanwhile, to prevent the Base ControlNet from being confused by different conditions, we propose to add *condition-specific LoRA* layers to every linear layer of the Base ControlNet. In this manner, different condition-specific LoRAs are responsible for capturing unique characteristics of corresponding tasks, and therefore the shared Base ControlNet can concentrate on the common knowledge of I2I generation. The whole schema is shown in Fig. 3(a).

Specifically, suppose we have $K$ distinct types of base conditions $\{\mathbf{c}^{(k)}\}_{k=1}^{K}$ with corresponding data subsets $\{\mathscr{D}^{(k)}\}_{k=1}^{K}$, and let $\mathcal{C}_{\theta}$ denote the Base ControlNet and $\mathcal{L}_{\psi^{(k)}}$ denote the condition-specific LoRA responsible for the $k^{\text{th}}$ condition. In this context, we propose to adapt the noise prediction network from Eq. (3) as follows:

$$\boldsymbol{\epsilon}_{\theta,\psi^{(k)}}(\mathbf{x}_t, \mathbf{c}^{(k)}) = \mathcal{D}(\mathcal{E}(\mathbf{x}_t), \mathcal{C}_{\theta,\psi^{(k)}}(\text{VAE}(\mathbf{c}^{(k)}))) \tag{4}$$

where $\mathcal{C}_{\theta,\psi^{(k)}} = \mathcal{C}_{\theta} \oplus \mathcal{L}_{\psi^{(k)}}$ refers to the Base ControlNet $\mathcal{C}_{\theta}$ equipped with the $k^{\text{th}}$ LoRA $\mathcal{L}_{\psi^{(k)}}$, and we use $\text{VAE}(\mathbf{c}^{(k)})$ as the condition embedding network to achieve faster training convergence (explained in Section 3.4). To optimize the Base ControlNet simultaneously on $K$ base conditions, we adapt Eq. (2) to the following loss function:

$$\mathscr{L}(\theta, \psi^{(1:K)}) = \sum_{k=1}^{K} \mathbb{E}_{(\mathbf{x}_0, \mathbf{c}^{(k)}) \sim \mathscr{D}^{(k)}} \left[ \mathbb{E}_{t \sim U(0,T), \boldsymbol{\epsilon} \sim \mathcal{N}(\mathbf{0}, \mathbf{I})} \left[ \left\| \boldsymbol{\epsilon} - \boldsymbol{\epsilon}_{\theta, \psi^{(k)}}(\mathbf{x}_t, \mathbf{c}^{(k)}) \right\|^2 \right] \right] \tag{5}$$

In practice, only a single condition is selected in each training batch and different conditions are iterated batch-wise, therefore all conditions can be optimized with an equal number of training iterations. For each batch, the LoRA layers corresponding to the current condition are switched on and updated, as shown in Fig. 3(a).

To ensure the effectiveness and generalizability of the Base ControlNet, the training process is conducted on 9 base conditions with millions of data (Qin et al., 2024) and takes around 6000 GPU hours. Although resource-consuming, this process paves the way for efficient adaptation to novel conditions as demonstrated in Section 3.3.

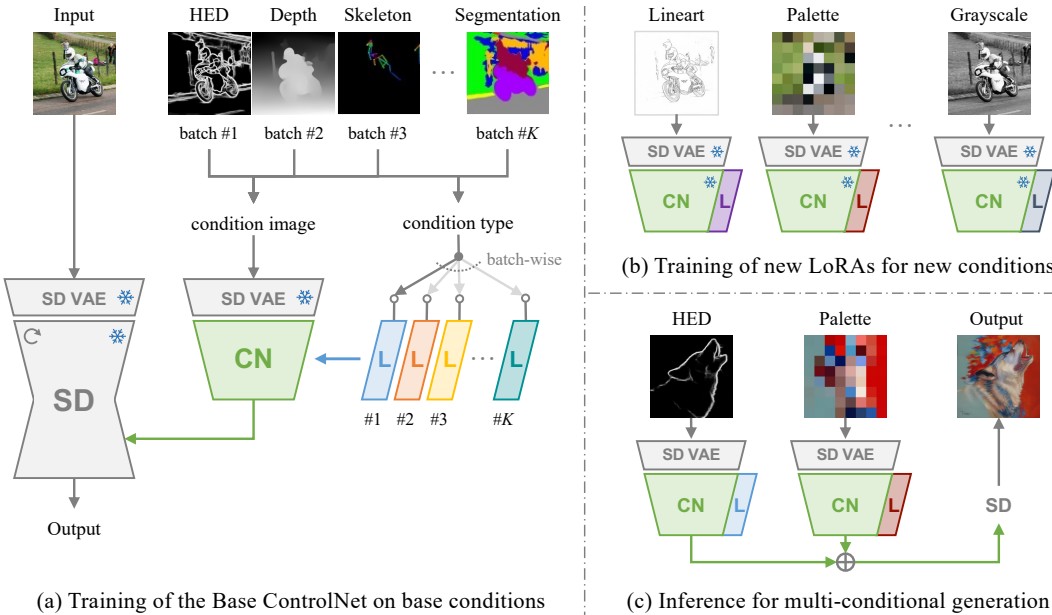

Figure 3: Training and inference of our CtrLoRA framework. "SD" denotes Stable Diffusion, "CN" denotes Base ControlNet, and "L"'s in different colors denote LoRAs for different conditions.

## 3.3 Efficient Adaptation to Novel Conditions

Since the well-trained Base ControlNet learns sufficient general knowledge of I2I generation, it can be efficiently adapted to new conditions via parameter-efficient fine-tuning. Similar to the LoRAs for the base conditions in Section 3.2, we can also train a new LoRA for any new condition while freezing the Base ControlNet, as shown in Fig. 3(b). As a result, there are only 37 million optimizable parameters when setting the LoRA rank as 128, a substantial reduction compared to 360 million parameters for full-parameter fine-tuning. Moreover, in most scenarios, as few as 1,000 data pairs and less than one hour of training on a single RTX 4090 GPU are sufficient for satisfactory results.

In addition, the LoRAs trained for different conditions can be composed for multi-conditional generation. Specifically, we can generate images that satisfy multiple conditions by summing up the outputs of the Base ControlNet equipped with corresponding LoRAs, as shown in Fig. 3(c).

## 3.4 Design of Condition Embedding Network

In the original ControlNet (Zhang et al., 2023), a simple convolutional network with random initialization is employed to map the condition image into an embedding, which is referred to as the *condition embedding network*. However, a randomly initialized network cannot extract any useful information from the condition image at the beginning of training and thus causes slow convergence.

To solve this problem, instead of a randomly initialized network, we propose to employ the pretrained VAE of Stable Diffusion (Rombach et al., 2022) as the condition embedding network, as shown in Fig. 3 and Eq. (4). For one thing, since the pretrained VAE has been proven to be powerful to represent and reconstruct an image (Rombach et al., 2022), it can already extract meaningful embedding from the condition image without extra learning. For another, since the Base ControlNet is initialized as a trainable copy of the Stable Diffusion encoder, the embedding space of the pretrained VAE seamlessly matches the initial input space of the Base ControlNet. In a word, compared to a randomly initialized network, using the pretrained VAE as the condition embedding network requires no extra effort to learn a suitable embedding space and therefore achieves much faster convergence. Besides, utilizing this method, the sudden convergence phenomenon associated with the original ControlNet is not observed anymore.

## 4 Experiments

### 4.1 Experiment Setup

**Datasets** To train the Base ControlNet, we employ a large-scale dataset, MultiGen-20M (Qin et al., 2024), which contains over 20 million image-condition pairs across 9 image-to-image tasks. To train the LoRAs for new conditions, we create multiple types of image-condition pairs based on COCO2017 (Lin et al., 2014) training set. For all quantitative evaluations, we employ COCO2017 validation set. Additionally, we use HazeWorld dataset (Xu et al., 2023) for dehazing task, Raindrop dataset (Qian et al., 2018) for de-raindrop, the dataset from Yang et al. (2020) for low-light image enhancement, and Danbooru2019 dataset (Branwen et al., 2019) for anime generation.

**Evaluation metrics** We use LPIPS (Zhang et al., 2018) to measure the faithfulness of the generated images to the condition images in two scenarios. For conditions including Canny, HED, Sketch, Depth, Normal, Segmentation, Skeleton, Lineart, and Densepose, the target is to generate images that match the condition images. Thus we re-extract the conditions from the generated images and compare them with the real condition images. For conditions including Outpainting, Inpainting, and Dehazing, the target is to generate high-fidelity images from degraded images. Therefore, we compare the generated images with the ground-truth images. Besides, we use FID score (Heusel et al., 2017) to evaluate the image quality.

**Implementation details** For a fair comparison with other methods, we use Stable Diffusion v1.5 in all experiments. We set the LoRA rank to 128 for each base task when training the Base ControlNet. The Base ControlNet is trained with AdamW optimizer (Loshchilov & Hutter, 2017) for 700k steps with a learning rate of $1 \times 10^{-5}$ and a batch size of 32, which takes around 6000 GPU hours on 8 RTX4090 GPUs. For all new conditions, the LoRA rank is set to 128. Besides, we

also fine-tune the normalization layers and the zero-convolutions. We use AdamW optimizer with a learning rate of $1\times10^{-5}$ and a batch size of 1. At this stage, only one GPU is needed, which is much more affordable than the requirements for training the original ControlNet. For sampling, we apply DDIM (Song et al., 2021a) sampler with 50 steps. The weight of classifier-free guidance (Ho & Salimans, 2022) is set to 7.5 and the strength of ControlNet is set to 1.0. We do not use any additional prompts or negative prompts for the quantitative evaluation.

## 4.2 COMPARISON WITH EXISTING METHODS

**Performance on base conditions** To demonstrate the capacity of the Base ControlNet, we evaluate its performance on the base conditions, as shown in Table 2 and Fig. 4. We compared the results to UniControl (Qin et al., 2024), a state-of-the-art method that trains a unified model to manage all base conditions, similar to our Base ControlNet. As can be seen, for base conditions, our base ControlNet performs on par with the state-of-the-art UniControl, demonstrating its robust fundamental capabilities. Furthermore, our base ControlNet can be easily and efficiently extended to new conditions, which is not straightforward using UniControl.

Table 2: Quantitative comparison on base conditions. Each cell represents "LPIPS↓ / FID↓".

|  | Canny | HED | Sketch | Depth | Normal | Segmentation | Skeleton | Outpainting | Bounding Box |
|---|---|---|---|---|---|---|---|---|---|
| UniControl | **0.273** / 18.58 | **0.176** / **13.97** | 0.391 / 24.95 | **0.216** / 21.29 | **0.319** / 24.90 | 0.467 / 22.02 | **0.129** / 53.64 | **0.527** / 14.10 | **0.292** / 26.65 |
| CtrLoRA (ours) | 0.388 / **16.65** | 0.251 / 14.75 | **0.288** / **19.17** | 0.222 / **19.34** | 0.329 / **18.67** | **0.465** / **21.13** | 0.132 / **51.40** | 0.549 / **13.96** | 0.315 / **23.95** |

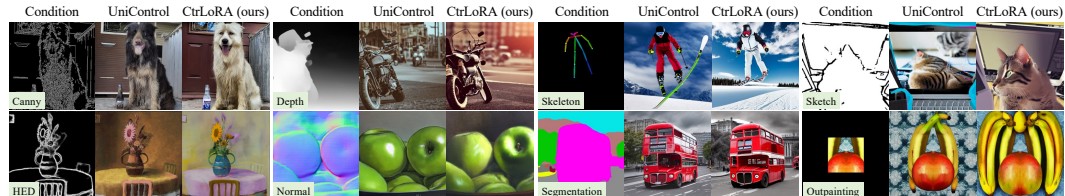

Figure 4: Visual comparison on base conditions.

Table 3: Quantitative comparison on new conditions. Each cell represents "LPIPS↓ / FID↓".

|  | Lineart | | Densepose | | Inpainting | | Dehazing | |
|---|---|---|---|---|---|---|---|---|
| # training images | 1k images | 100k images | 1k images | 100k images | 1k images | 100k images | 1k images | 100k images |
| ControlNet | 0.622 / 22.29 | 0.264 / 14.10 | 0.367 / 36.80 | 0.140 / 33.36 | 0.785 / 22.09 | 0.465 / 12.79 | 0.758 / 54.07 | 0.348 / 22.85 |
| ControlNet-LITE | 0.623 / 23.00 | 0.267 / 15.24 | 0.368 / 36.70 | 0.152 / 34.51 | 0.785 / 22.88 | 0.530 / 14.86 | 0.761 / 60.42 | 0.409 / 27.54 |
| ControlNet-XS | 0.623 / 22.36 | **0.245** / 15.33 | 0.368 / 36.46 | 0.148 / **32.50** | 0.784 / 22.81 | 0.503 / 13.37 | 0.761 / 59.59 | 0.397 / 25.83 |
| CtrLoRA (ours) | **0.305** / **16.12** | 0.247 / **13.47** | **0.159** / **35.18** | **0.126** / 32.80 | **0.326** / **9.972** | **0.246** / **8.214** | **0.255** / **15.44** | **0.178** / **10.55** |

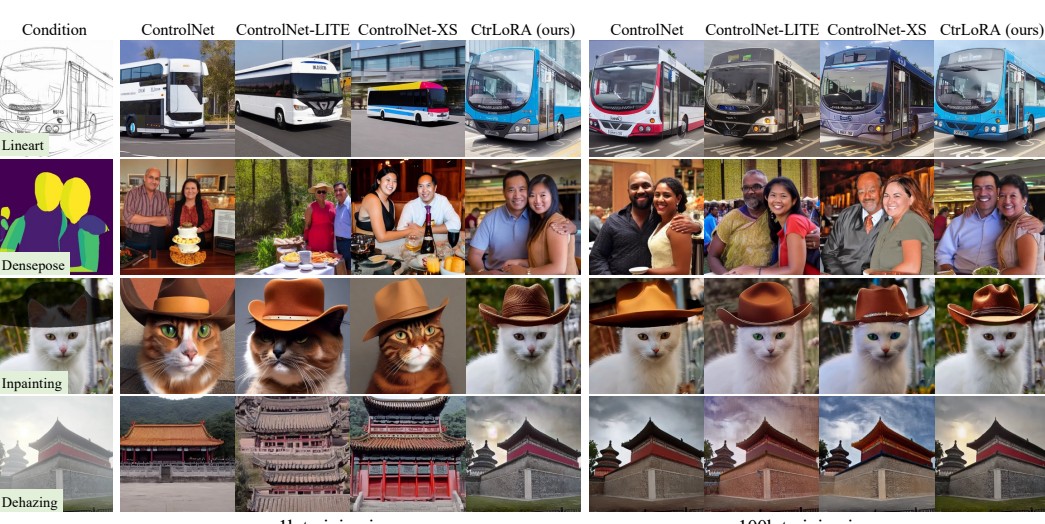

Figure 5: Visual comparison on new conditions.

**Adaptation to new conditions** For new conditions, we compare our method with Control-Net (Zhang et al., 2023), ControlNet-LITE (Zhang et al., 2023), and ControlNet-XS (Zavadski et al., 2023). The latter two are lightweight alternatives to ControlNet, aiming to optimize the network architecture and accelerate the training process. To evaluate the data efficiency and scalability, we conduct experiments on 1k and 100k training images respectively, as shown in Table 3 and Fig. 5. With a limited training set (1k), CtrLoRA consistently outperforms the competitors by a large margin, highlighting its superiority in quickly adapting to new conditions. With a large training set (100k), CtrLoRA achieves better or comparable results. In summary, regarding the adaptation to new conditions, our CtrLoRA is not only highly data-efficient but can also achieve satisfactory performance as the data scale increases.

**Convergence rate** We visualize the results with respect to training steps and plot the convergence curve in Fig. 6. As can be seen, our CtrLoRA starts to follow the condition after just 500 training steps, while the other methods take more than 10,000 steps to reach convergence.

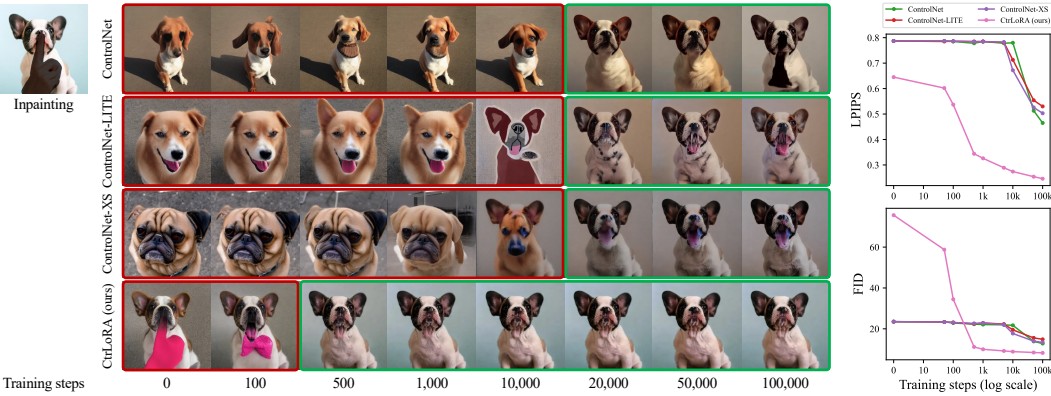

Figure 6: Visual comparison of convergence rate.

## 4.3 ABLATION STUDY

**Effect of each component** We validate the effect of our components by starting with the original ControlNet and adding our proposed components one by one, resulting in three incremental settings (A-C) in Table 4. In setting (A) of Table 4 and Fig. 7, we evaluate the effect of using the pretrained VAE as the condition embedding network, as proposed in Section 3.4. As can be seen, using the pretrained VAE improves both LPIPS and FID score as well as speeding up the training convergence. In setting (B), we further switch the initialization of ControlNet from Stable Diffusion to a well-trained Base ControlNet and perform full-parameter fine-tuning, in order to validate the generalizability of our Base ControlNet. As shown, our Base ControlNet can be more quickly adapted to new conditions with a limited training set (1k images) and still achieves leading performance with a large training set (100k images). This result demonstrates that our Base ControlNet learns sufficient general knowledge of I2I generation and indeed helps the adaptation to novel conditions. At last in setting (C), we replace the full-parameter training with condition-specific LoRAs, which represents the complete implementation of our method. As shown, although the LoRAs reduce the optimizable parameters by 90%, it does not lose much performance and maintains the second-best performance in most situations, demonstrating the effectiveness and efficiency of our CtrLoRA framework.

Table 4: Effect of the proposed components. Each cell represents "LPIPS↓ / FID↓".

| | Lineart | | Densepose | | Inpainting | | Dehazing | |
|---|---|---|---|---|---|---|---|---|
| # training images | 1k images | 100k images | 1k images | 100k images | 1k images | 100k images | 1k images | 100k images |
| (O) ControlNet | 0.622 / 22.29 | 0.264 / 14.10 | 0.367 / 36.80 | 0.140 / 33.36 | 0.785 / 22.09 | 0.465 / 12.79 | 0.758 / 54.07 | 0.348 / 22.85 |
| (A) + Pretrained VAE | 0.393 / 20.04 | 0.248 / 13.85 | 0.217 / 36.48 | 0.129 / 33.40 | 0.489 / 19.05 | 0.257 / 8.941 | 0.350 / 27.07 | 0.180 / 11.07 |
| (B) + Base ControlNet | **0.300 / 14.25** | **0.228 / 12.71** | **0.138 / 34.02** | 0.130 / 32.56 | **0.296 / 9.450** | 0.253 / 8.265 | **0.221 / 13.47** | **0.160 / 10.08** |
| (C) + LoRA (full CtrLoRA) | 0.305 / 16.12 | 0.247 / 13.47 | 0.159 / 35.18 | **0.126** / 32.80 | 0.326 / 9.972 | **0.246 / 8.214** | 0.255 / 15.44 | 0.178 / 10.55 |

*The best and second results are highlighted in **boldface** and underlined respectively.*

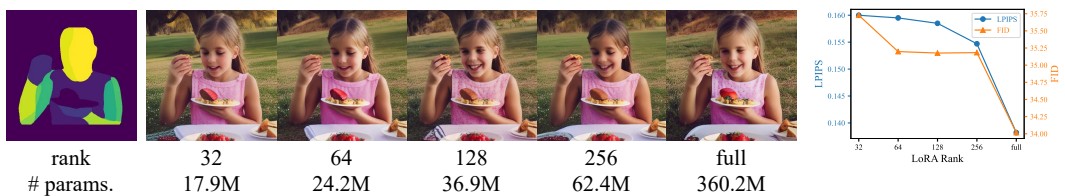

Figure 7: Convergence rate comparison between ControlNet and setting (A).

**Effect of LoRA rank**  We evaluate the CtrLoRA performance with LoRA ranks of 32, 64, 128, and 256, and we also evaluate the full-parameter training as the upper bound. As shown in Fig. 8, LPIPS improves as the rank increases, while FID score plateaus at the rank of 64. To balance the performance and number of optimizable parameters, we choose a rank of 128 for all conditions.

**Effect of training set size**  We train our CtrLoRA respectively on datasets containing 1k, 3k, 5k, 10k, and 50k images, running 5 epochs for each dataset size. As shown in Fig. 9, both LPIPS and FID improve as the dataset size increases. Nevertheless, in our practice for most new conditions, a small amount of training data (1k images) is generally sufficient for satisfactory visual perception.

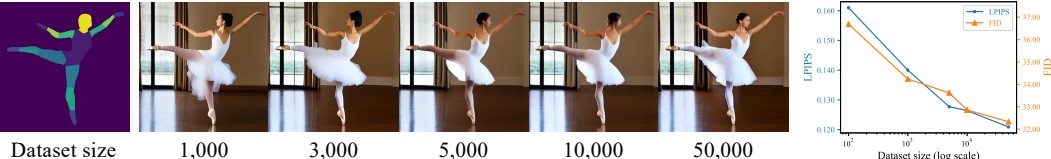

Figure 8: Effect of LoRA rank.

Figure 9: Effect of training set size.

## 4.4 OTHER EXPERIMENTS

**More novel conditions**  We provide visual results of more novel conditions including Palette, Lineart with color prompt, Pixel, De-raindrop, Low-light image enhancement, and Illusion in Fig. 10. Despite the significant differences among these conditions, our method yields decent results across all of them, which demonstrates the generalizability of our CtrLoRA to a wide range of conditions.

**Integration into community models**  Our CtrLoRA can be directly applied to the community models based on Stable Diffusion 1.5. In Fig. 11(a), we integrate our CtrLoRA into four community models with markedly distinct styles. The results exhibit different styles but remain consistent with the given conditions, suggesting that our method can be flexibly used as a plug-and-play module without extra training.

**Combine multiple conditions**  By equipping the Base ControlNet with different LoRAs and summing their outputs, we can perform multi-conditional generation without extra training. The weight assigned to each condition can be manually adjusted to control its effect on the final result, with an equal weight of 1 typically sufficient in most scenarios. As shown in Fig. 11(b), our CtrLoRA can generate visually appealing images that comply with both conditions simultaneously.

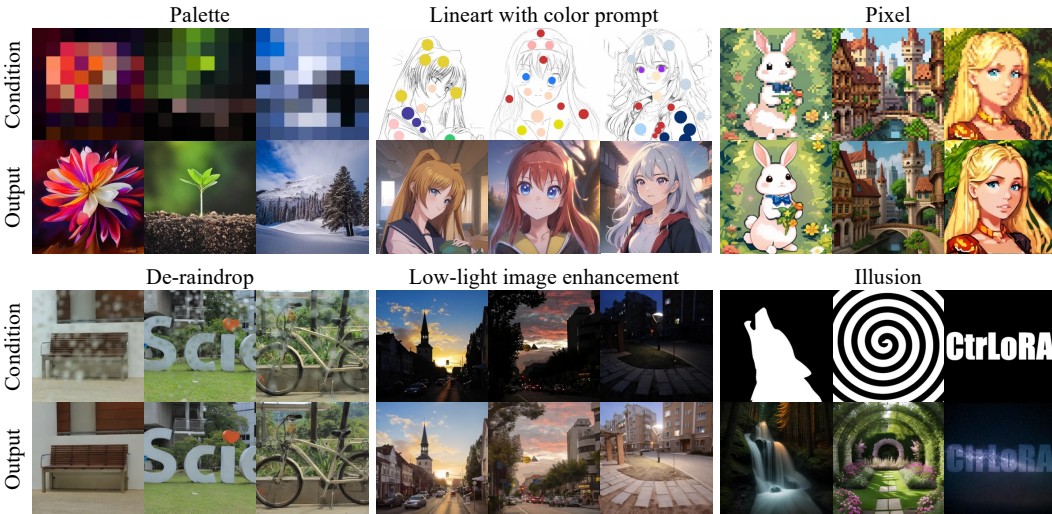

Figure 10: Visual results of our CtrLoRA for various novel conditions.

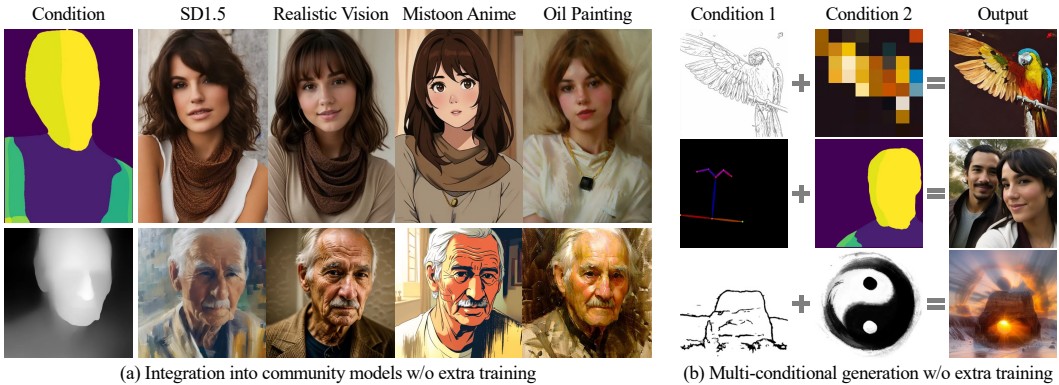

(a) Integration into community models w/o extra training     (b) Multi-conditional generation w/o extra training

Figure 11: Without extra training, a well-trained CtrLoRA can be directly integrated into various community models and combined for multi-conditional generation.

## 5 CONCLUSION AND LIMITATIONS

In this paper, we propose CtrLoRA, a framework aimed at developing a controllable generation model for any new condition with minimal data and resources. In this framework, we first train a Base ControlNet along with condition-specific LoRAs to capture the common knowledge of image-to-image generation, and then adapt it to new conditions by training new LoRAs. Compared to ControlNet, our approach significantly reduces the requirement for data and computational resources and greatly accelerates training convergence. Furthermore, the trained models can be seamlessly integrated into community models and combined for multi-conditional generation without further training. By lowering the development threshold, we hope our research will encourage more people to join the community and facilitate the development of controllable image generation.

We empirically found that color-related conditions, such as Palette and Lineart with color prompts, tend to converge more slowly than conditions involving only spatial relationships. This phenomenon seems to be a common issue that not only appears in our methods but also in other ControlNet-based competitors. We speculate this issue might originate from the capabilities of the network architectures, specifically the architectures of VAE, UNet-based Stable Diffusion, and ControlNet. To enhance the capabilities of our framework, it is worth developing our CtrLoRA using more advanced DiT-based (Peebles & Xie, 2023) backbones such as Stable Diffusion V3 (Esser et al., 2024) and Flux.1, which we leave for future work.

ACKNOWLEDGMENTS

This work is partially supported by the National Natural Science Foundation of China (No. 62461160331 and 62406311), the Postdoctoral Fellowship Program of CPSF (No. GZB20230774), and the Innovation Funding of ICT, CAS (No. E361020).

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

# APPENDIX

## A  SUDDEN CONVERGENCE PHENOMENON

Sudden convergence is an intriguing phenomenon observed in the original ControlNet (Zhang et al., 2023), where the generated images suddenly converge to match the condition image in only 33 training steps (6100 to 6133). The original paper attributes this phenomenon to the use of zero-convolutions. However, we found that the *condition embedding network*, originally implemented as a randomly initialized convolutional network, also contributes to this phenomenon. As elaborated in Section 3.4, we propose using the pretrained VAE as the condition embedding network to accelerate the training convergence, the effect of which is shown in Table 4 and Fig. 7 (Setting (A)) of the main paper. Furthermore, we found that our design also alleviates the sudden convergence phenomenon. Specifically, in Fig. 12, we display the generated images every 25 training steps from 2,000 to 3,000, given the same condition. As can be seen, the results oscillate between compliance and non-compliance with the given condition, which corresponds to an oscillation near the local minimum rather than a sudden convergence. This result implies that an improper design of the condition embedding network is indeed one cause of the sudden convergence phenomenon, and using the pretrained VAE as in our paper is an effective solution.

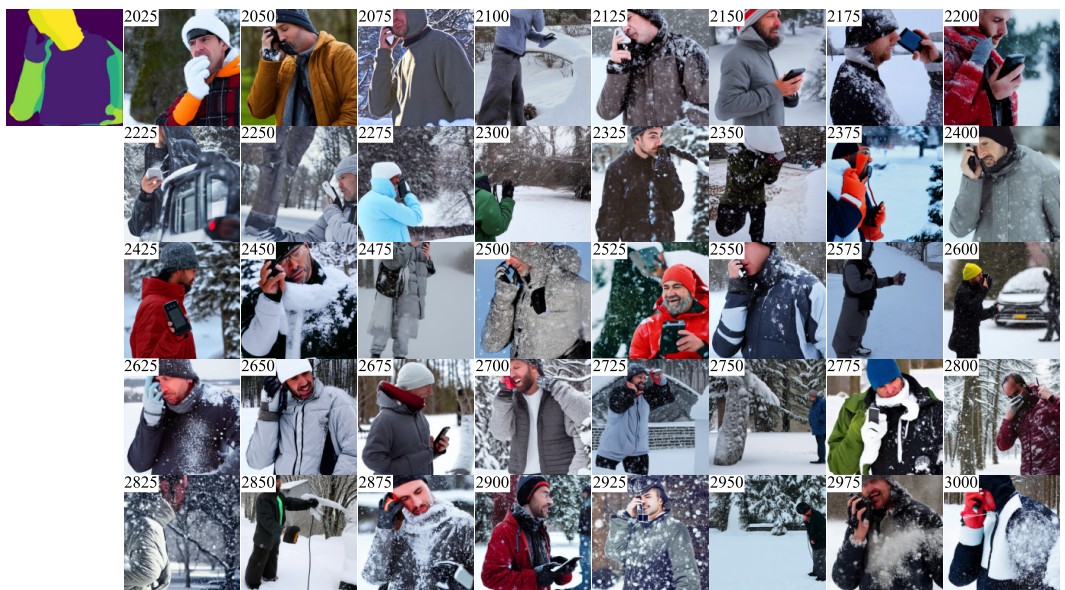

Figure 12: Generated images between 2,000 and 3,000 training steps.

## B  ADDITIONAL DISCUSSION ON RELATED EFFICIENT METHODS

T2I-Adapter (Mou et al., 2024) and SCEdit (Jiang et al., 2024) are two efficient alternatives of ControlNet that mainly focus on decreasing the model size. However, the data and GPU resources needed to train these models are still beyond the reach of ordinary users. For example, T2I-Adapter is trained on 164k to 600k images with 4 V100 GPUs for around 3 days, and SCEdit is trained on 600k images with 16 A100 GPUs. On the contrary, our method can achieve satisfactory performance by training on about 1,000 images with a single RTX 4090 GPU within 1 hour, while keeping the model size comparable or even smaller, thereby significantly lowering the cost for ordinary users to create their customized ControlNets.

## C  MORE QUANTITATIVE RESULTS

**Controllable generation benchmark**  To establish a comprehensive benchmark on controllable image-to-image generation, we compare our method with several representative community models, including official models of ControlNet (Zhang et al., 2023) and T2I-Adapter (Mou et al., 2024). As shown in Table 5 and Table 6, our CtrLoRA outperforms fully trained ControlNet and T2I-Adapter for both base and new conditions. The visual comparisons are shown in Fig. 17 and Fig. 18.

Table 5: Benchmark for base conditions. Each cell represents "LPIPS↓ / FID↓".

|  | Canny | Depth | Segmentation | Skeleton |
|---|---|---|---|---|
| ControlNet | 0.438 / 17.80 | 0.232 / 20.09 | 0.488 / **20.83** | 0.134 / **50.79** |
| T2I-Adapter | 0.447 / 18.45 | 0.305 / 23.81 | 0.636 / 21.59 | 0.137 / 52.92 |
| UniControl | **0.273** / 18.58 | **0.216** / 21.29 | 0.467 / 22.02 | **0.129** / 53.64 |
| CtrLoRA (ours) | 0.388 / **16.65** | 0.222 / **19.34** | **0.465** / 21.13 | 0.132 / 51.40 |

Table 6: Benchmark for novel conditions. Each cell represents "LPIPS↓ / FID↓".

|  | Lineart | Densepose | Inpainting | Dehazing |
|---|---|---|---|---|
| ControlNet | 0.254 / 15.04 | 0.140 / 33.36[†] | 0.465 / 12.79[†] | 0.348 / 22.85[†] |
| T2I-Adapter | 0.498 / 20.53 | - | - | - |
| CtrLoRA (ours)[‡] | **0.247 / 13.47** | **0.126 / 32.80** | **0.246 / 8.214** | **0.178 / 10.55** |

[†] *ControlNet for Densepose, Inpainting, and Dehazing are trained by ourselves on 100k images.*
[‡] *Our CtrLoRAs are trained on 100k images for each novel condition.*

**Necessity of our Base ControlNet**  To demonstrate the necessity and adaptability of our Base ControlNet, we compare our method to directly fine-tuning a pretrained ControlNet and UniControl Qin et al. (2024). We also compare with ControlLoRA (Hecong, 2023) that directly trains LoRAs with conditional input on Stable Diffusion. We limit the training set to 1,000 images to assess the effectiveness of each method in quickly adapting to new conditions. As shown in Table 7, our method significantly outperforms these methods in adapting to new conditions, showing the effectiveness of our Base ControlNet and the potential of our idea to learn the general knowledge of I2I generation.

Table 7: Fine-tuning with limited data (1,000 images). Each cell represents "LPIPS↓ / FID↓".

|  | Lineart | Densepose | Inpainting | Dehazing |
|---|---|---|---|---|
| ControlNet (canny) + LoRA | 0.356 / 16.74 | 0.198 / 36.14 | 0.602 / 17.63 | 0.618 / 51.55 |
| UniControl + LoRA | 0.316 / 17.05 | 0.164 / 41.20 | 0.558 / 15.84 | 0.508 / 37.83 |
| ControlLoRA | 0.362 / 17.28 | 0.295 / **32.37** | 0.614 / 21.92 | 0.472 / 41.96 |
| CtrLoRA (ours) | **0.305 / 16.12** | **0.159** / 35.18 | **0.326 / 9.972** | **0.255 / 15.44** |

**Effect of the number of base conditions**  To investigate how the number of base conditions affects the adaptability of our Base ControlNet to learn new conditions, we train three Base ControlNets on 3, 6, and 9 base conditions respectively and fine-tune them to new conditions. As shown in Table 8, the overall performance on new conditions gets better when more base conditions are included to train the Base ControlNet, demonstrating that the Base ControlNet can extract better general knowledge from more conditions.

Table 8: Effect of the number of base conditions. Each cell represents "LPIPS↓ / FID↓".

| # Base conditions | Lineart | Densepose | Inpainting | Dehazing |
|---|---|---|---|---|
| 3 | 0.348 / 15.71 | 0.161 / 35.63 | 0.461 / 14.63 | 0.312 / 23.16 |
| 6 | 0.324 / 15.59 | 0.159 / **35.25** | 0.343 / **10.73** | 0.262 / 17.14 |
| 9 | **0.307 / 15.06** | **0.157** / 35.31 | **0.337** / 10.84 | **0.248 / 16.23** |

*3 base conditions include Canny, Depth, Skeleton*
*6 base conditions include Canny, Depth, Skeleton, Segmentation, Bounding Box, Outpainting*
*9 base conditions include Canny, Depth, Skeleton, Segmentation, Bounding Box, Outpainting, HED, Sketch, Normal*

## D MORE VISUAL RESULTS

In this part, we provide more visual results of our CtrLoRA, including Fig. 14 for base conditions, Fig. 15 for new conditions, Fig. 16 for multi-conditional generation, Fig. 17 and Fig. 18 for visual comparison with officially released models of ControlNet (Zhang et al., 2023) and T2I-Adapter (Mou et al., 2024).

## E CTRLORA FOR STYLE TRANSFER

As demonstrated in Section 4.4, our CtrLoRA can be flexibly integrated into stylized community models and perform multi-conditional generation without extra training. These features open the door to exploring more creative and exciting applications such as style transfer. Given a reference image, we first choose a stylized model to specify the target style. Then, we combine the well-trained LoRAs for Palette and Lineart to control the color and shape of the output image, according to the reference. The overall process is shown in Fig. 13 and exemplar results are shown in Fig. 19.

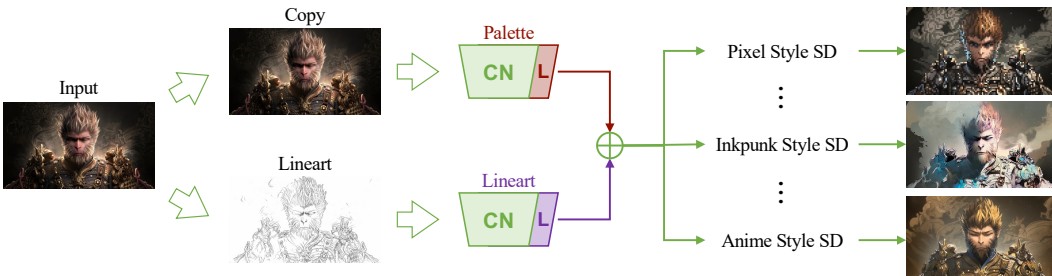

Figure 13: The overall process of style transfer.

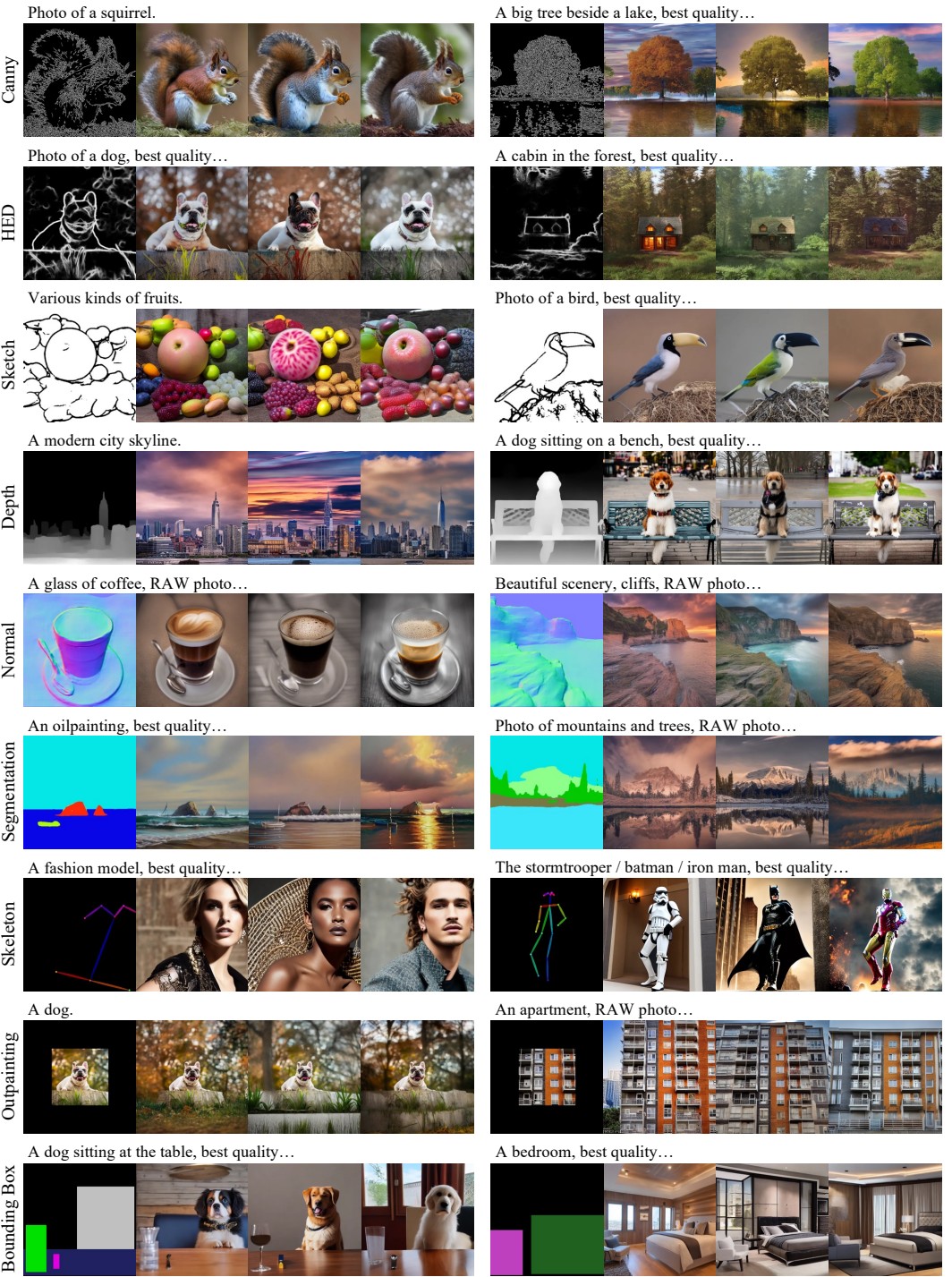

Figure 14: More visual results of base conditions.

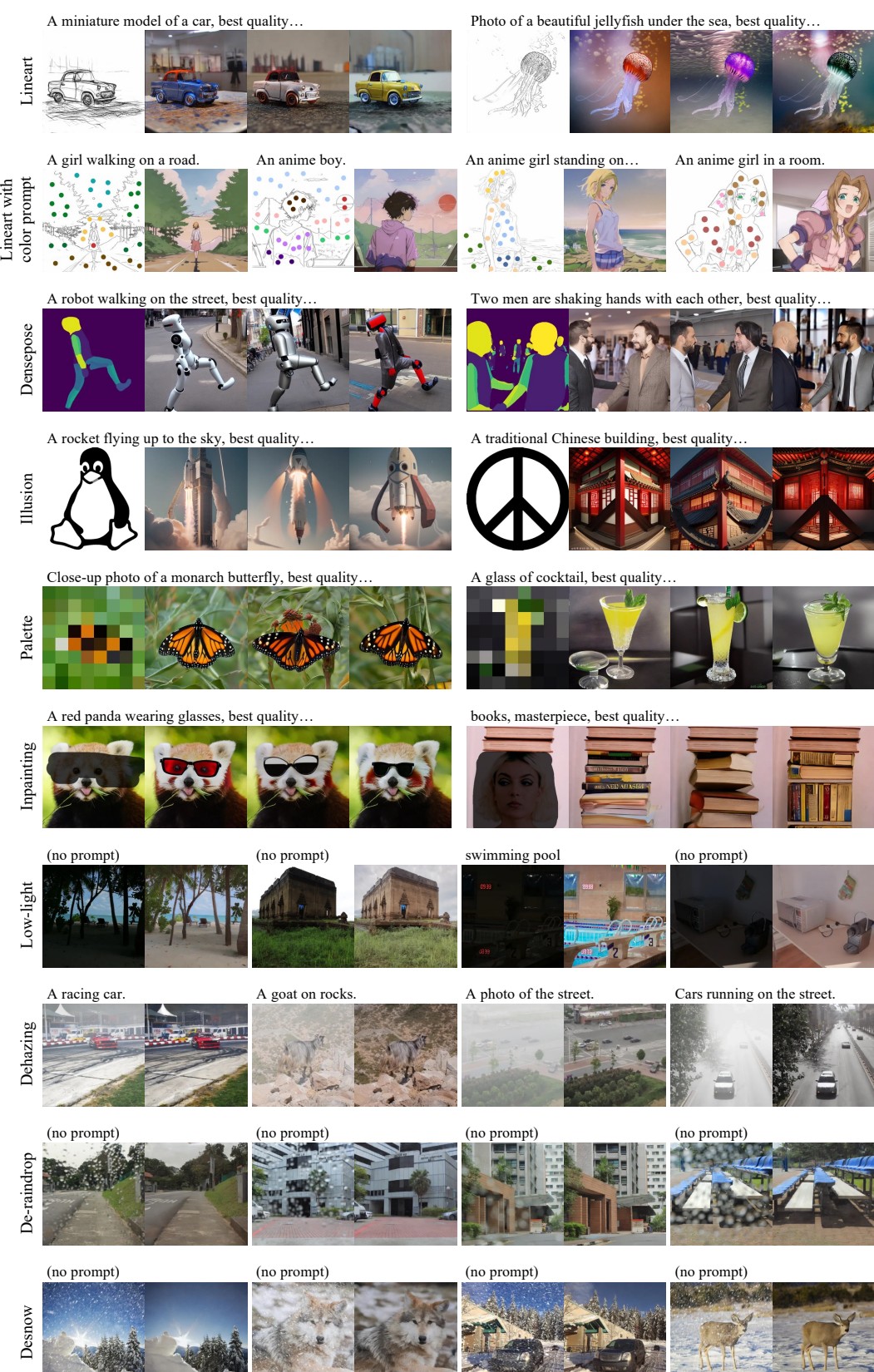

Figure 15: More visual results of new conditions.

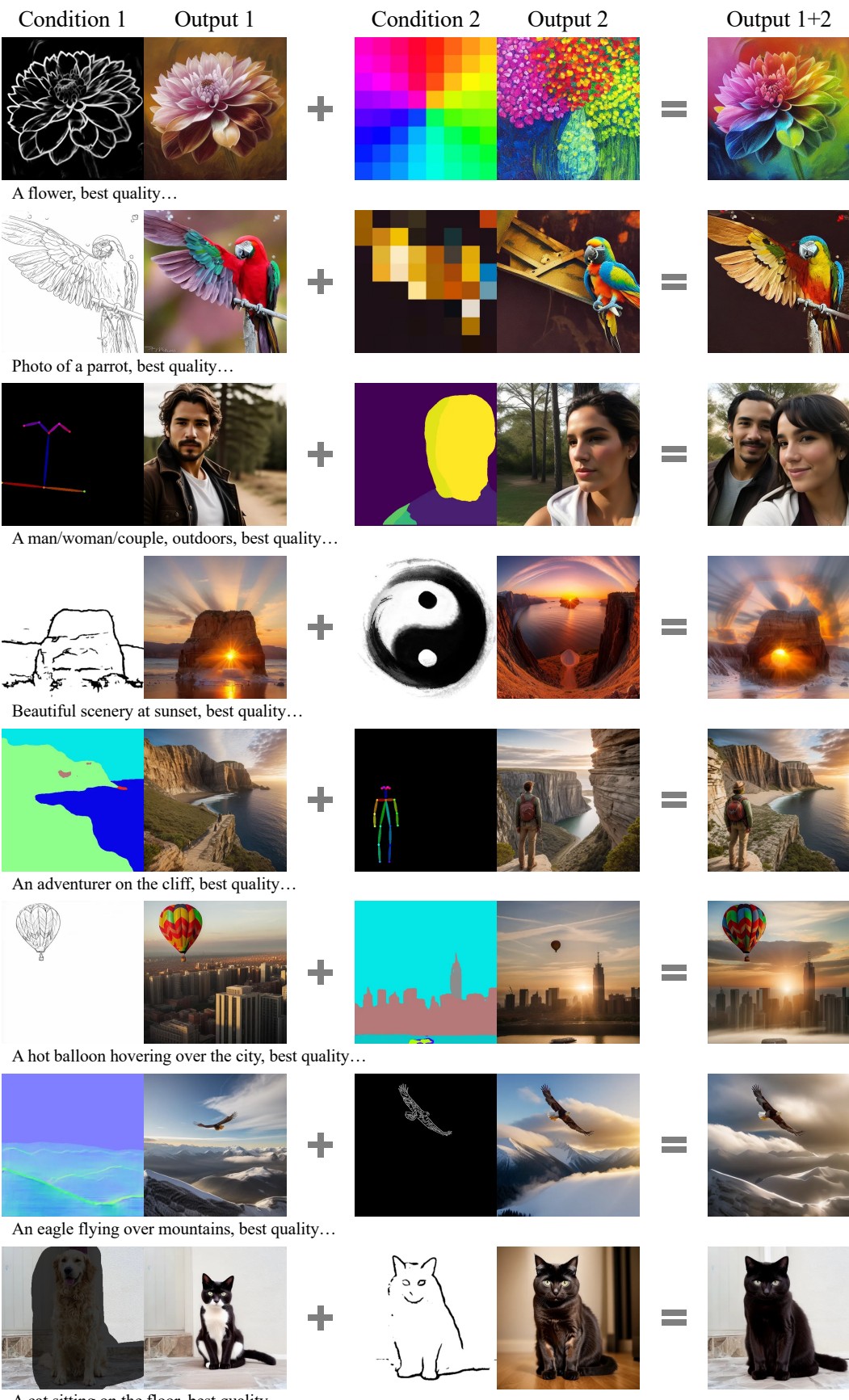

Figure 16: More visual results of multi-conditional generation.

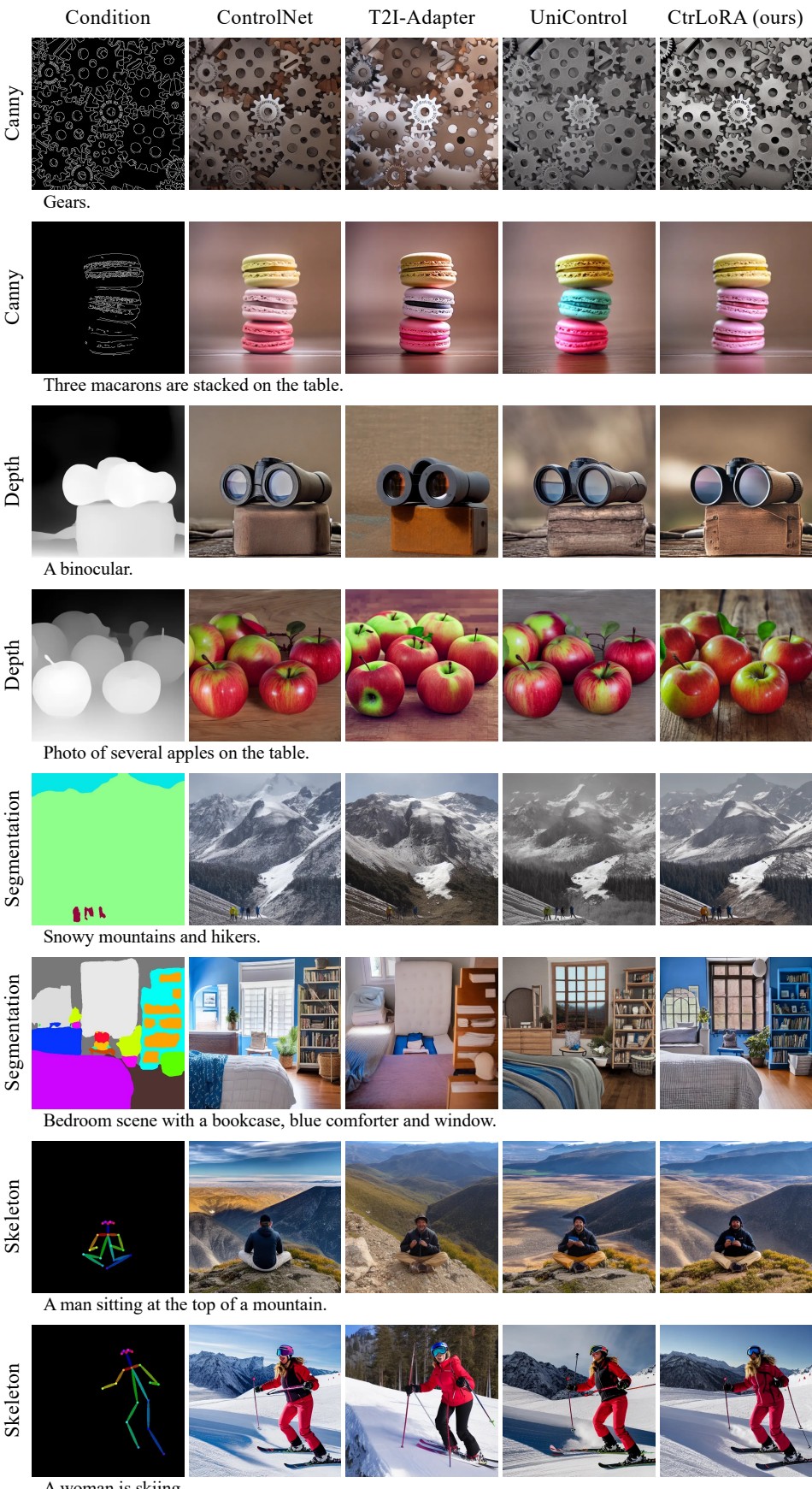

Figure 17: Visual comparison with fully trained community models on base conditions.

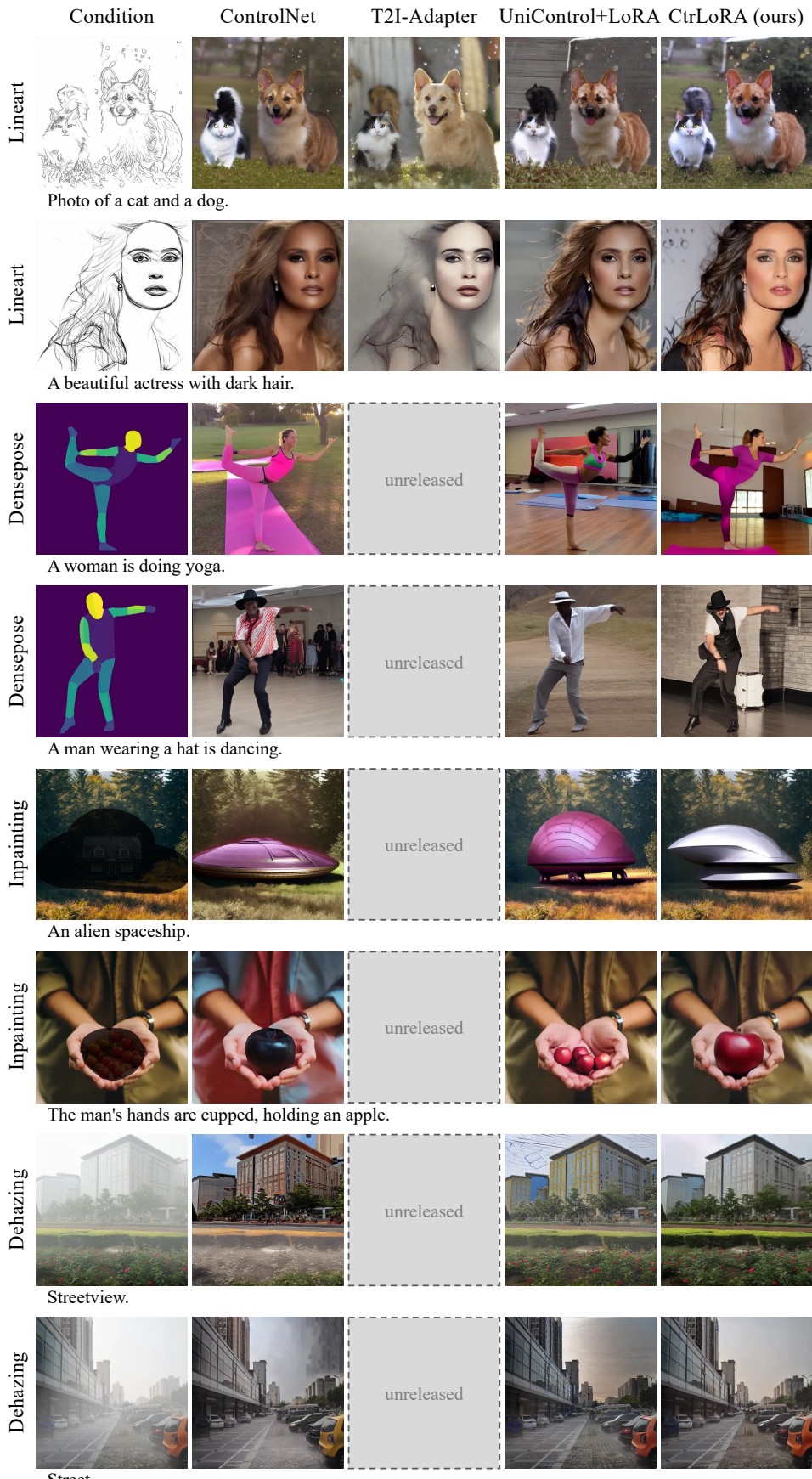

Figure 18: Visual comparison with fully trained community models on new conditions.

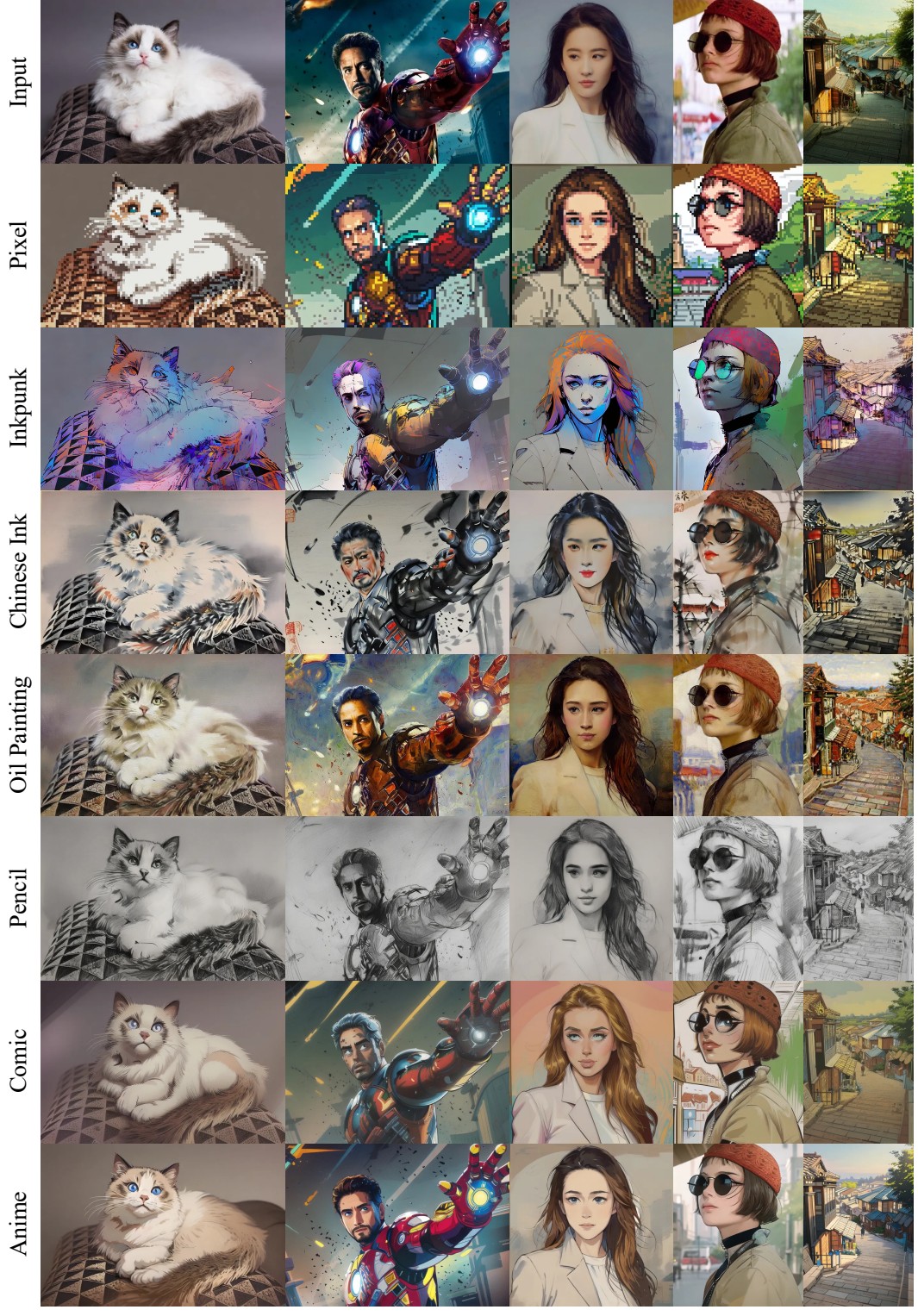

Figure 19: Results of style transfer.

## F  PROMPTS AND MODELS FOR VISUAL RESULTS

Since this paper mainly focuses on I2I generation, we omit the text prompts in the main paper for simplicity. For a comprehensive presentation, here we provide the prompts and the base models corresponding to the visual results of the main paper, as shown in Table 9.

Table 9: Prompts and base models for visual results in the main paper.

| Figure | Stable Diffusion | Prompt |
|---|---|---|
| Fig. 4 | SD1.5 | A dirty dog sits on the front patio of a home. |
| | SD1.5 | A parked motorcycle sitting on a dirty road. |
| | SD1.5 | A man riding skis while flying through the air. |
| | SD1.5 | A cat observing a computer screen next to a laptop and a cordless phone. |
| | SD1.5 | The painting is of a vase of flowers on a table. |
| | SD1.5 | A large white bowl of many green apples. |
| | SD1.5 | The red, double decker bus is driving past other buses. |
| | SD1.5 | There are bananas around another piece of fruit. |
| Fig. 5 | SD1.5 | A passenger bus pulling up to the side of a street, best quality… |
| | SD1.5 | A man and a women posing next to one another in front of a table, RAW photo… |
| | SD1.5 | a cat wearing a brown cowboy hat, best quality… |
| | SD1.5 | An ancient Chinese building. |
| Fig. 6 | SD1.5 | a dog |
| Fig. 7 | SD1.5 | Baseball game, RAW photo… |
| Fig. 8 | SD1.5 | A girl is eating dessert at the table, picnic, RAW photo… |
| Fig. 9 | SD1.5 | A girl wearing white dress is dancing ballet |
| Fig. 10 | SD1.5 | a flower, best quality… |
| | SD1.5 | A close up photo of a green seedling breaks out of the ground, RAW photo… |
| | SD1.5 | mountain and trees in winter, best quality… |
| | Mistoon Anime | an anime girl, best quality |
| | Mistoon Anime | an anime girl, outdoors, best quality |
| | Mistoon Anime | an anime girl, best quality |
| | Mistoon Anime | a cute rabbit |
| | Realistic Vision | city |
| | Mistoon Anime | a girl |
| | SD1.5 | A bench. |
| | SD1.5 | (no prompt) |
| | SD1.5 | (no prompt) |
| | Realistic Vision | streetview, best quality… |
| | Realistic Vision | streetview, best quality… |
| | Realistic Vision | buildings, best quality… |
| | Dreamshaper | forest at night, best quality… |
| | Realistic Vision | garden |
| | Realistic Vision | sky with stars, RAW photo… |
| Fig. 11(a) | SD1.5 | A girl with brown hair and a necklace wearing a cowl-neck shirt, best quality… |
| | Realistic Vision | A girl with brown hair and a necklace wearing a cowl-neck shirt, best quality… |
| | Mistoon Anime | A girl with brown hair and a necklace wearing a cowl-neck shirt, best quality… |
| | Oil Painting | A girl with brown hair and a necklace wearing a cowl-neck shirt, best quality… |
| | SD1.5 | An old man, best quality… |
| | Realistic Vision | An old man, best quality… |
| | Mistoon Anime | An old man, best quality… |
| | Oil Painting | An old man, best quality… |
| Fig. 11(b) | SD1.5 | Photo of a parrot, best quality |
| | Realistic Vision | a couple, outdoors, best quality… |
| | Realistic Vision | Beautiful scenery at sunset, best quality… |

