# OpenReview forum: "CtrLoRA: An Extensible and Efficient Framework for Controllable Image Generation"
_ICLR.cc/2025/Conference — ICLR 2025 Poster_

### Official Review · Reviewer_97kn · 2024-10-23

**Soundness:** 3
**Presentation:** 3
**Contribution:** 3
**Rating:** 6
**Confidence:** 5

**Summary:**

In this paper, the authors propose a CtrloRA framework. This framework starts by training a basic ControlNet that handles various image conditions efficiently. With this trained network, one can quickly fine-tune it to adapt to new conditions using a task-specific LoRA—specifically, fine-tuning requires only 1,000 paired images and less than an hour on a single GPU. The experimental results confirm that this method greatly speeds up the training process for new image conditions. Based on these impressive findings, I recommend a weak acceptance. However, there are some unclear points and missing experiments in the paper (see the Question section), and my final decision will depend on the authors' responses to these issues.

**Strengths:**

The CtrloRA framework introduced in this paper allows users to quickly and efficiently fine-tune the ControlNet to new image conditions, with minimal resource consumption. The experimental results validate the effectiveness of this method. Additionally, the paper is well-structured and clearly written.

**Weaknesses:**

There are some unclear points and missing experiments in the paper (see the Question section), and my final decision will depend on the authors' responses to these issues.

**Questions:**

1. Consider specifying 1-2 new image conditions and key metrics (e.g., adaptation speed, data efficiency, performance) for comparing UniControl [1] fine-tuning to CtrLoRA. This would provide a clear, focused comparison.
2. Additional baselines are required for each base image condition. Comparisons should be made with a fully trained ControlNet, which has been trained exclusively under a single image condition, to establish a more comprehensive benchmark.
3. Similarly, for the new condition, it is essential to compare the performance of CtrLora against ControlNet when ControlNet has been fully trained on a single modality. This will provide a clearer understanding of their relative efficiencies.
4. It would be beneficial to explore how the number of image conditions used during the training of the base ControlNet affects its ability to learn new conditions. Insights into the scalability and adaptability of the base network could prove crucial for future applications.
5. I have noted that CtrloRA can perform low-level image enhancement tasks, such as low-light image enhancement. Could the authors demonstrate how CtrloRA performs in comparison to other diffusion models for low-light image enhancement? This could highlight potential advantages or unique features of CtrloRA in practical applications.

[1] UniControl: A Unified Diffusion Model for Controllable Visual Generation In the Wild.

---

> ### Author Response · Authors · 2024-11-20
> **Response to Reviewer 97kn (Part 1/3)**
>
> Thank you very much for your constructive suggestions for the experiments.
>
> &nbsp;
>
> > **Q1:**  Consider specifying 1-2 new image conditions and key metrics (e.g., adaptation speed, data efficiency, performance) for comparing UniControl [1] fine-tuning to CtrLoRA. This would provide a clear, focused comparison.
>
> |                           | Inpainting-1k                             | Inpainting-100k                           | Dehazing-1k                               | Dehazing-100k                             |
> | ------------------------- | ----------------------------------------- | ----------------------------------------- | ----------------------------------------- | ----------------------------------------- |
> |                           | LPIPS↓ / FID↓                             | LPIPS↓ / FID↓                             | LPIPS↓ / FID↓                             | LPIPS↓ / FID↓                             |
> | ControlNet (canny) + LoRA | 0.602 / 17.63                             | 0.412 / 11.22                             | 0.618 / 51.55                             | 0.320 / 19.96                             |
> | UniControl + LoRA         | $\underline{0.558}$ / $\underline{15.84}$ | $\underline{0.337}$ / $\underline{9.580}$ | $\underline{0.508}$ / $\underline{37.83}$ | $\underline{0.271}$ / $\underline{17.06}$ |
> | CtrLoRA (ours)            | **0.326** / **9.972**                     | **0.246** / **8.214**                     | **0.255** / **15.44**                     | **0.178** / **10.55**                     |
>
> Although UniControl [1] trains a unified model on multiple conditions, its delicate design makes it not straightforward to be quickly extended to new conditions. On the contrary, our fine-tuning stage keeps consistent with the pre-training strategy of our Base ControlNet, and therefore the adaptation to new conditions is natural and efficient. As can be seen from the above comparison, our CtrLoRA significantly outperforms UniControl, demonstrating the superior adaptability of our method to new conditions. It can also be observed that our CtrLoRA trained on 1000 images even surpass UniControl trained on 100,000 images, demonstrating the data efficiency of our method.

---

> ### Author Response · Authors · 2024-11-20
> **Response to Reviewer 97kn (Part 2/3)**
>
> > **Q2:** Additional baselines are required for each base image condition. Comparisons should be made with a fully trained ControlNet, which has been trained exclusively under a single image condition, to establish a more comprehensive benchmark.
> >
> > **Q3:** Similarly, for the new condition, it is essential to compare the performance of CtrLora against ControlNet when ControlNet has been fully trained on a single modality. This will provide a clearer understanding of their relative efficiencies.
>
> Thanks for this valuable suggestion. Below, we compare our method with multiple community models, including fully trained ControlNet from the community. It can be seen that our CtrLoRA outperforms fully trained ControlNet for both base and new conditions. (The ControlNet for Densepose, Inpainting and Dehazing are trained by ourselves with 100k images.)
>
> &nbsp;
>
> **Base conditions:**
>
> |                         | Canny                           | Depth                           | Segmentation                    | Skeleton                    |
> | ----------------------- | ------------------------------- | ------------------------------- | ------------------------------- | --------------------------- |
> |                         | LPIPS↓ / FID↓                   | LPIPS↓ / FID↓                   | LPIPS↓ / FID↓                   | LPIPS↓ / FID↓               |
> | ControlNet (community)  | 0.438 / $\underline{17.80}$     | 0.232 / $\underline{20.09}$     | 0.488 / **20.83**               | 0.134 / **50.79**           |
> | T2I-Adapter (community) | 0.447 / 18.45                   | 0.305 / 23.81                   | 0.636 / 21.59                   | 0.137 / 52.92 |
> | UniControl (community)  | **0.273** / 18.58               | **0.216** / 21.29               | $\underline{0.467}$ / 22.02     | **0.129** / 53.64           |
> | CtrLoRA (ours)          | $\underline{0.388}$ / **16.65** | $\underline{0.222}$ / **19.34** | **0.465** / $\underline{21.13}$ | $\underline{0.132}$ / $\underline{51.40}$ |
>
> &nbsp;
>
> **New conditions:**
>
> |                                 | Lineart                         | Densepose                       | Inpainting                                | Dehazing                                  |
> | ------------------------------- | ------------------------------- | ------------------------------- | ----------------------------------------- | ----------------------------------------- |
> |                                 | LPIPS↓ / FID↓                   | LPIPS↓ / FID↓                   | LPIPS↓ / FID↓                             | LPIPS↓ / FID↓                             |
> | ControlNet (community)          | 0.254 / 15.04                   | 0.140 / $\underline{33.36}$                   | 0.465 / 12.79                             | 0.348 / 22.85                             |
> | T2I-Adapter (community)         | 0.498 / 20.53                   | -                               | -                                         | -                                         |
> | UniControl + LoRA (100k images) | **0.224** / $\underline{14.26}$ | **0.124** / 36.51 | $\underline{0.337}$ / $\underline{9.580}$ | $\underline{0.271}$ / $\underline{17.06}$ |
> | CtrLoRA (ours) (100k images)    | $\underline{0.247}$ / **13.47** | $\underline{0.126}$ / **32.80** | **0.246** / **8.214**                     | **0.178** / **10.55**                     |
>
> ---
>
> &nbsp;
>
> > **Q4:** It would be beneficial to explore how the number of image conditions used during the training of the base ControlNet affects its ability to learn new conditions. Insights into the scalability and adaptability of the base network could prove crucial for future applications.
>
> *We are currently running experiments on the number of base conditions and will present the results as soon as possible*.

---

> > ### Author Response · Authors · 2024-11-24
> > **Updated Response for Q4**
> >
> > > **Q4:** It would be beneficial to explore how the number of image conditions used during the training of the base ControlNet affects its ability to learn new conditions. Insights into the scalability and adaptability of the base network could prove crucial for future applications.
> >
> > We train three Base ControlNets on 3, 6, and 9 base conditions respectively and fine-tune them to new conditions.
> > As shown below, the overall performance on the new conditions gets better when more base conditions are included to train the Base ControlNet, demonstrating that the Base ControlNet can extract better general knowledge from more conditions.
> >
> > | \# Base conditions |                  Lineart                  |             Densepose              |              Inpainting              |                 Dehazing                  |
> > | :----------------: | :---------------------------------------: | :--------------------------------: | :----------------------------------: | :---------------------------------------: |
> > |         3          |               0.348 / 15.71               |           0.161 / 35.63            |            0.461 / 14.63             |               0.312 / 23.16               |
> > |         6          | $\underline{0.324}$ / $\underline{15.59}$ |  $\underline{0.159}$ / **35.25**   |   $\underline{0.343}$ / **10.73**    | $\underline{0.262}$ / $\underline{17.14}$ |
> > |         9          |      **0.307** / **15.06**      | **0.157** / $\underline{35.31}$ | **0.337** / $\underline{10.84}$ |      **0.248** / **16.23**      |
> >
> > *3 base conditions include canny, depth, skeleton*
> >
> > *6 base conditions include canny, depth, skeleton, segmentation, bounding box, outpainting*
> >
> > *9 base conditions include canny, depth, skeleton, segmentation, bounding box, outpainting, hed, sketch, normal*

---

> ### Author Response · Authors · 2024-11-20
> **Response to Reviewer 97kn (Part 3/3)**
>
> > **Q5:** I have noted that CtrloRA can perform low-level image enhancement tasks, such as low-light image enhancement. Could the authors demonstrate how CtrloRA performs in comparison to other diffusion models for low-light image enhancement? This could highlight potential advantages or unique features of CtrloRA in practical applications.
>
> Since we have no special design for these low-level tasks, it can be almost sure that we cannot perform better than methods specialized in these tasks. In this period, we train and compare a state-of-the-art method RetinexFormer [2] for low-light image enhancement. As can be seen, our CtrLoRA lags behind the state-of-the-art performance.
>
> |                | LPIPS↓ | PSNR↑   |
> | -------------- | ------ | ------- |
> | RetinexFormer  | 0.2064 | 19.5137 |
> | CtrLoRA (ours) | 0.2912 | 15.8184 |
>
> However, note that the main purpose of training CtrLoRA for low-level image enhancement tasks is to prove the generalizability of our method to various novel conditions. Although far from state-of-the-art, the results are visually satisfactory and significantly better than related competitors (see the dehazing performance in Table 3 of the paper).
>
> ---
>
> &nbsp;
>
> [1] Qin, Can, et al. "UniControl: A Unified Diffusion Model for Controllable Visual Generation In the Wild." Advances in Neural Information Processing Systems 36 (2024).
>
> [2] Cai, Yuanhao, et al. "Retinexformer: One-stage retinex-based transformer for low-light image enhancement." Proceedings of the IEEE/CVF International Conference on Computer Vision. 2023.

---

> ### Author Response · Authors · 2024-11-24
> **Looking forward to more discussions**
>
> We highly appreciate your constructive feedback.
>
> We have carefully responded to each of your questions and revised our paper accordingly. The revision details are listed in ["Paper Updates"](https://openreview.net/forum?id=3Gga05Jdmj&noteId=isfjp1umdb) at the top of this page.
> Besides, we've clarified our motivation and contribution in the ["General Response"](https://openreview.net/forum?id=3Gga05Jdmj&noteId=v3zsgz6v6e) at the top of this page.
>
> We look forward to your reply and welcome any discussion on unclear points regarding our paper and our response.

---

### Official Review · Reviewer_YjAj · 2024-10-29

**Soundness:** 3
**Presentation:** 3
**Contribution:** 2
**Rating:** 6
**Confidence:** 5

**Summary:**

This paper draws on the idea of combining a base model with PEFT (Parameter-Efficient Fine-Tuning) for controllable generation. It trains a Base ControlNet obtained through several condition-specific training processes, and then fine-tunes it with a small amount of data for newly introduced conditions to obtain different condition-specific LoRAs. This approach improves the efficiency of training new condition generators at a lower cost.

**Strengths:**

- To address the high cost of separately training different models for conditional generation tasks, this paper proposes a training method that transitions from a base controlnet model to a lightly fine-tuned lora model. This approach ensures generation quality while achieving a faster convergence rate.

- The paper shows many analyses of the proposed method and presents the results generated for a total of more than a dozen conditions.

- The paper is well structured and easy to follow.

**Weaknesses:**

- The paper primarily aims to improve the training efficiency of all kinds of conditional models, hence it employs a series of LoRAs to train the newly introduced conditions based on the "Base ControlNet". However, there is relatively little comparison and discussion of existing methods that efficiently train ControlNet, such as T2I-Adapter, ControlLoRA, and SCEdit.

- There currently exists a viable **controlnet-union** model, which can handle different conditions using a single model. This may be a higher-level representation of the training of the "Base ControlNet" model discussed in the paper. On the other hand, the use of LoRA for fine-tuning is relatively straightforward and has been implemented in previous community works, such as ControlLoRA. In comparison, the overall innovativeness of the paper is limited.

- The paper does not discuss how many conditions to use or how to select conditions for training the "Base ControlNet" to achieve optimal knowledge transfer effects.

**Questions:**

- Regarding the discussion of "Adaptation to new conditions," while training a comparison method from scratch with a small amount of data may indeed result in slow convergence, what would be the results if we used a pre-trained conditional model (analogous to possessing a Base ControlNet) for fine-tuning?

- I'm curious about the performance between a pre-trained controlnet model available in the community and a model trained using proposed "Base + LoRA" with same conditions.

---

> ### Author Response · Authors · 2024-11-20
> **Response to Reviewer YjAj (Part 1/3)**
>
> We highly appreciate your constructive comments.
>
> &nbsp;
>
> > **W1:** The paper primarily aims to improve the training efficiency of all kinds of conditional models, hence it employs a series of LoRAs to train the newly introduced conditions based on the "Base ControlNet". However, there is relatively little comparison and discussion of existing methods that efficiently train ControlNet, such as T2I-Adapter, ControlLoRA, and SCEdit.
>
> Existing efficient methods such as T2I-Adapter [1] and SCEdit [3] mainly focus on decreasing the model sizes, but the data and GPU resources needed to train these models are still beyond the reach of ordinary users. For example, T2I-Adapter is trained on 164k~600k images with 4 V100 GPUs for around 3 days, and SCEdit is trained on 600k images with 16 A100 GPUs. On the contrary, our method can achieve satisfactory performance by training on ~1000 images with a single RTX 4090 GPU within 1 hour, while keeping the model sizes comparable to or even smaller than T2I-Adapter and SCEdit, thereby greatly lowering the cost for ordinary users to create their customized ControlNets.
>
> As for ControlLoRA [2], which also employs LoRA for conditional generation, it suffers from poor performance with a small amount of training data. Below we show the results of training ControlLoRA on 1000 images. It is clear that our method significantly outperforms ControlLoRA. We will further discuss the core difference between our method and ControlLoRA in the response to W2.2 below.
>
> |                | Lineart               | Densepose         | Inpainting            | Dehazing              |
> | -------------- | --------------------- | ----------------- | --------------------- | --------------------- |
> |                | LPIPS↓ / FID↓         | LPIPS↓ / FID↓     | LPIPS↓ / FID↓         | LPIPS↓ / FID↓         |
> | ControlLoRA    | 0.362 / 17.28         | 0.295 / **32.37** | 0.614 / 21.92         | 0.472 / 41.96         |
> | CtrLoRA (ours) | **0.305** / **16.12** | **0.159** / 35.18 | **0.326** / **9.972** | **0.255** / **15.44** |
>
> *We will add these discussions in the revised paper*.
>
> ---
>
> &nbsp;
>
> > **W2.1:** There currently exists a viable controlnet-union model, which can handle different conditions using a single model. This may be a higher-level representation of the training of the "Base ControlNet" model discussed in the paper.
>
> Thanks for pointing out controlnet-union [4]. We found that it was released on July 2, thus it should be considered as a concurrent work. This method is similar to UniControl [5] and Uni-ControlNet [6] which manage multiple conditions within a unified model. However, while these methods only emphasize **union**, our work also emphasizes **adaptation to new conditions at a substantially low cost**. Thus we are solving distinct problems with different techniques.
>
> ---
>
> &nbsp;
>
> > **W2.2:** the use of LoRA for fine-tuning is relatively straightforward and has been implemented in previous community works, such as ControlLoRA. In comparison, the overall innovativeness of the paper is limited.
>
> Using LoRA for new conditions "seems to be straightforward", which has been already tried by the community model ControlLoRA [2] as you mentioned. However, LoRA itself is not the challenge; **the real challenge is how to make a new LoRA perform well with limited data (1000 images in our paper)**, as it is difficult for an ordinary user to collect a large customized dataset. This challenge is not straightforward to solve and existing methods cannot handle it well including ControlLoRA. To this end, we propose to train a Base ControlNet with shifting strategy to capture the general knowledge of I2I generation. With our well trained Base ControlNet, 1000 data samples is sufficient to learn a LoRA for a new condition with satisfactory results.
>
>
>
> Compared to ControlLoRA, which directly uses LoRA to fine-tune Stable Diffusion, our method emphasizes the importance of general I2I knowledge (the Base ControlNet). The results in the response to W1 demonstrate that our method is significantly superior to ControlLoRA. We believe the emphasis on the necessity of a Base ControlNet is one of our main contributions and significantly distinguishes our method from ControlLoRA.
>
> ---
>
> &nbsp;
>
> > **W3:** The paper does not discuss how many conditions to use or how to select conditions for training the "Base ControlNet" to achieve optimal knowledge transfer effects.
>
> *We are currently running experiments on the number of base conditions and will present the results as soon as possible*.

---

> > ### Author Response · Authors · 2024-11-24
> > **Updated Response for W3**
> >
> > > **W3:** The paper does not discuss how many conditions to use or how to select conditions for training the "Base ControlNet" to achieve optimal knowledge transfer effects.
> >
> > We train three Base ControlNets on 3, 6, and 9 base conditions respectively and fine-tune them to new conditions.
> > As shown below, the overall performance on the new conditions gets better when more base conditions are included to train the Base ControlNet, demonstrating that the Base ControlNet can extract better general knowledge from more conditions.
> >
> > | \# Base conditions |                  Lineart                  |             Densepose              |              Inpainting              |                 Dehazing                  |
> > | :----------------: | :---------------------------------------: | :--------------------------------: | :----------------------------------: | :---------------------------------------: |
> > |         3          |               0.348 / 15.71               |           0.161 / 35.63            |            0.461 / 14.63             |               0.312 / 23.16               |
> > |         6          | $\underline{0.324}$ / $\underline{15.59}$ |  $\underline{0.159}$ / **35.25**   |   $\underline{0.343}$ / **10.73**    | $\underline{0.262}$ / $\underline{17.14}$ |
> > |         9          |      **0.307** / **15.06**      | **0.157** / $\underline{35.31}$ | **0.337** / $\underline{10.84}$ |      **0.248** / **16.23**      |
> >
> > *3 base conditions include canny, depth, skeleton*
> >
> > *6 base conditions include canny, depth, skeleton, segmentation, bounding box, outpainting*
> >
> > *9 base conditions include canny, depth, skeleton, segmentation, bounding box, outpainting, hed, sketch, normal*
> >
> > &nbsp;
> >
> > Regretfully, since training a Base ControlNet requires a lot of time and devices, we are not able to ablate the selection of base conditions during the rebuttal period. However, we have an intuitive opinion: without any prior knowledge, all base conditions should be viewed equally. For example, one particular selection of base conditions may be optimal for a new condition A, but sub-optimal for another new condition B. Generally speaking, we cannot predict what kind of new conditions the users are dealing with; therefore, it is a natural choice to treat all base conditions equally.

---

> ### Author Response · Authors · 2024-11-20
> **Response to Reviewer YjAj (Part 2/3)**
>
> > **Q1:** Regarding the discussion of "Adaptation to new conditions," while training a comparison method from scratch with a small amount of data may indeed result in slow convergence, what would be the results if we used a pre-trained conditional model (analogous to possessing a Base ControlNet) for fine-tuning?
>
> A straightforward manner is to fine-tune a pretrained ControlNet or UniControl [5]. However, both are less effective than our method. As discussed at line 173 of our paper, a pre-trained ControlNet is extensively trained to fit a particular condition, and therefore not general enough to efficiently adapt to different conditions. For UniControl, as discussed at line 161 of our paper, although it is trained on multiple conditions, its delicate design makes it not straightforward to be quickly extended to new conditions. On the contrary, our fine-tuning stage keeps consistent with the pre-training strategy of our Base ControlNet, and therefore the adaptation to new conditions is natural and efficient.
>
> Below we add the comparison to directly fine-tune a pre-trained ControlNet and UniControl on 1000 images. As can be seen, our CtrLoRA significantly outperforms these methods when adapting to new conditions, demonstrating the effectiveness of our Base ControlNet and the potential of our idea to learn the general knowledge of I2I generation.
>
> |                           | Lineart                     | Densepose                       | Inpainting                                | Dehazing                                  |
> | ------------------------- | --------------------------- | ------------------------------- | ----------------------------------------- | ----------------------------------------- |
> |                           | LPIPS↓ / FID↓               | LPIPS↓ / FID↓                   | LPIPS↓ / FID↓                             | LPIPS↓ / FID↓                             |
> | ControlNet (canny) + LoRA | 0.356 / $\underline{16.74}$ | 0.198 / $\underline{36.14}$               | 0.602 / 17.63                             | 0.618 / 51.55                             |
> | UniControl + LoRA         | $\underline{0.316}$ / 17.05 | $\underline{0.164}$ / 41.20     | $\underline{0.558}$ / $\underline{15.84}$ | $\underline{0.508}$ / $\underline{37.83}$ |
> | CtrLoRA (ours)            | **0.305** / **16.12**       | **0.159** / **35.18** | **0.326** / **9.972**                     | **0.255** / **15.44**                     |

---

> ### Author Response · Authors · 2024-11-20
> **Response to Reviewer YjAj (Part 3/3)**
>
> > **Q2:** I'm curious about the performance between a pre-trained controlnet model available in the community and a model trained using proposed "Base + LoRA" with same conditions.
>
> Thanks for this valuable suggestion. Below, we compare our method with multiple community models. As for base conditions, our CtrLoRA achieves comparable performance to the state-of-the-art method UniControl [5] and outperforms the rest of the competitors. As for novel conditions, our CtrLoRA performs better than the competitors in most cases. (The ControlNet for Densepose, Inpainting and Dehazing are trained by ourselves with 100k images.)
>
> &nbsp;
>
> **Base conditions:**
>
> |                         | Canny                           | Depth                           | Segmentation                    | Skeleton                    |
> | ----------------------- | ------------------------------- | ------------------------------- | ------------------------------- | --------------------------- |
> |                         | LPIPS↓ / FID↓                   | LPIPS↓ / FID↓                   | LPIPS↓ / FID↓                   | LPIPS↓ / FID↓               |
> | ControlNet (community)  | 0.438 / $\underline{17.80}$     | 0.232 / $\underline{20.09}$     | 0.488 / **20.83**               | 0.134 / **50.79**           |
> | T2I-Adapter (community) | 0.447 / 18.45                   | 0.305 / 23.81                   | 0.636 / 21.59                   | 0.137 / 52.92 |
> | UniControl (community)  | **0.273** / 18.58               | **0.216** / 21.29               | $\underline{0.467}$ / 22.02     | **0.129** / 53.64           |
> | CtrLoRA (ours)          | $\underline{0.388}$ / **16.65** | $\underline{0.222}$ / **19.34** | **0.465** / $\underline{21.13}$ | $\underline{0.132}$ / $\underline{51.40}$ |
>
> &nbsp;
>
> **New conditions:**
>
> |                                 | Lineart                         | Densepose                       | Inpainting                                | Dehazing                                  |
> | ------------------------------- | ------------------------------- | ------------------------------- | ----------------------------------------- | ----------------------------------------- |
> |                                 | LPIPS↓ / FID↓                   | LPIPS↓ / FID↓                   | LPIPS↓ / FID↓                             | LPIPS↓ / FID↓                             |
> | ControlNet (community)          | 0.254 / 15.04                   | 0.140 / $\underline{33.36}$                   | 0.465 / 12.79                             | 0.348 / 22.85                             |
> | T2I-Adapter (community)         | 0.498 / 20.53                   | -                               | -                                         | -                                         |
> | UniControl + LoRA (100k images) | **0.224** / $\underline{14.26}$ | **0.124** / 36.51 | $\underline{0.337}$ / $\underline{9.580}$ | $\underline{0.271}$ / $\underline{17.06}$ |
> | CtrLoRA (ours) (100k images)    | $\underline{0.247}$ / **13.47** | $\underline{0.126}$ / **32.80** | **0.246** / **8.214**                     | **0.178** / **10.55**                     |
>
> ---
>
> &nbsp;
>
> [1] Mou, Chong, et al. "T2i-adapter: Learning adapters to dig out more controllable ability for text-to-image diffusion models." Proceedings of the AAAI Conference on Artificial Intelligence. Vol. 38. No. 5. 2024.
>
> [2] Wu, Hecong. "ControlLoRA: A Lightweight Neural Network To Control Stable Diffusion Spatial Information." GitHub.
>
> [3] Jiang, Zeyinzi, et al. "Scedit: Efficient and controllable image diffusion generation via skip connection editing." Proceedings of the IEEE/CVF Conference on Computer Vision and Pattern Recognition. 2024.
>
> [4] xinsir, et al. "controlnet-union-sdxl-1.0." Hugging Face.
>
> [5] Qin, Can, et al. "UniControl: A Unified Diffusion Model for Controllable Visual Generation In the Wild." Advances in Neural Information Processing Systems 36 (2024).

---

> ### Author Response · Authors · 2024-11-24
> **Looking forward to more discussions**
>
> We highly appreciate your constructive feedback.
>
> We have carefully responded to each of your questions and revised our paper accordingly. The revision details are listed in ["Paper Updates"](https://openreview.net/forum?id=3Gga05Jdmj&noteId=isfjp1umdb) at the top of this page. Besides, we've clarified our motivation and contribution in the ["General Response"](https://openreview.net/forum?id=3Gga05Jdmj&noteId=v3zsgz6v6e) at the top of this page.
>
> We look forward to your reply and welcome any discussion on unclear points regarding our paper and our response.

---

> > ### Comment · Reviewer_YjAj · 2024-11-26
> >
> > Thank you for your response.
> >
> > Regarding the response to Q2: I suggest including some visualized results, which would allow for more intuitive and thorough comparisons. From the previous responses, I have already understood the importance of I2I knowledge learning and gained an intuitive understanding of the transfer effects based on quantitative evaluation metrics. However, through Fig. 15, I still cannot fully assess whether the generated results demonstrate comparable transfer capabilities.
> >
> > Regarding the response to W3: Does the presence of similar tasks in the base directly facilitate the transfer of similar tasks, and how does this impact other tasks? Can some analysis be conducted based on the existing results?
> >
> > I will consider adjusting my score if the above issues can be addressed.

---

> > > ### Author Response · Authors · 2024-11-26
> > >
> > > > Regarding the response to Q2: I suggest including some visualized results, which would allow for more intuitive and thorough comparisons.
> > >
> > > Thank you for the suggestion. We have uploaded a new revision that includes visual results corresponding to **Q2** (Figure 18 & Figure 19 in the Appendix).
> > >
> > >
> > >
> > > ---
> > >
> > > &nbsp;
> > >
> > >
> > >
> > > > Regarding the response to W3: Does the presence of similar tasks in the base directly facilitate the transfer of similar tasks, and how does this impact other tasks? Can some analysis be conducted based on the existing results?
> > >
> > > This is a very fundamental and profound question. *We copy the table from "Updated Response for W3" for clearer presentation.*
> > >
> > > | \# Base conditions |                  Lineart                  |            Densepose            |           Inpainting            |                 Dehazing                  |
> > > | :----------------: | :---------------------------------------: | :-----------------------------: | :-----------------------------: | :---------------------------------------: |
> > > |         3          |               0.348 / 15.71               |          0.161 / 35.63          |          0.461 / 14.63          |               0.312 / 23.16               |
> > > |         6          | $\underline{0.324}$ / $\underline{15.59}$ | $\underline{0.159}$ / **35.25** | $\underline{0.343}$ / **10.73** | $\underline{0.262}$ / $\underline{17.14}$ |
> > > |         9          |           **0.307** / **15.06**           | **0.157** / $\underline{35.31}$ | **0.337** / $\underline{10.84}$ |           **0.248** / **16.23**           |
> > >
> > > *3 base conditions include canny, depth, skeleton*
> > >
> > > *6 base conditions include canny, depth, skeleton, segmentation, bounding box, outpainting*
> > >
> > > *9 base conditions include canny, depth, skeleton, segmentation, bounding box, outpainting, hed, sketch, normal*
> > >
> > > &nbsp;
> > >
> > > In this paper, we design a training scheme to let the Base ControlNet learn general I2I knowledge from a set of base conditions/tasks. However, if the number of the base conditions is small (e.g., 3 in the above table), we suppose that the Base ControlNet tends to learn the specific knowledge of the base conditions, and may indeed facilitate the transfer to new similar conditions.
> > >
> > > In the following, we analyze two representative novel conditions:
> > >
> > > + **Dehazing:** none of the base conditions/tasks is similar to dehazing
> > > + **Lineart:** all three above base sets contain **canny** that is similar to lineart
> > >
> > > Regarding **dehazing**, its performance significantly improves as the number of base conditions increases, even in the absence of similar conditions/tasks. This phenomenon demonstrates that our Base ControlNet indeed learns more useful common I2I knowledge with more base conditions.
> > >
> > > Regarding **lineart**, we suppose that some extent of particular ability/knowledge learned from **canny** may facilitate the learning of lineart, even when there are only three base conditions. Therefore, we can see the performance of lineart does not increase as fast as dehazing, maybe because a part of ability has already been learned from canny. Nevertheless, the performance still grows when more base conditions are included, which demonstrates that the increase of common knowledge can continue to improve the transfer.
> > >
> > > In summary, we can conclude that the more base conditions are included, the more common I2I knowledge can be learned by the Base ControlNet. Besides, the learning of new conditions may be facilitated if there are similar base conditions, but the common knowledge still takes effect.

---

> > > > ### Comment · Reviewer_YjAj · 2024-11-27
> > > >
> > > > The author provided a clear answer to my concern, and I will adjust my rating.

---

### Official Review · Reviewer_GM4E · 2024-11-01

**Soundness:** 3
**Presentation:** 2
**Contribution:** 2
**Rating:** 6
**Confidence:** 4

**Summary:**

This paper proposes CtrlLoRA, a two-stage parameter-efficient fine-tuning pipeline, to ease the original ControlNet's computation burden in terms of different conditions. The authors evaluate CtrlLoRA through extensive experiments by both the quality and the computation efficiency.

**Strengths:**

1. This paper focus on an important problem, extending ControlNet to a lightweight manner.
2. Experimental results are impressive, especially the convergence experiment.

**Weaknesses:**

1. In line 70, the authors state that ControlNet with Canny edges requires 3 million images over 600 GPU hours for one condition. In contrast, line 244 indicates that Base ControlNet necessitates millions of images for 6000 GPU hours for 9 conditions. Although it is not fair enough, but it implies that the proposed method does not significantly reduce the computational burden.

2. In line 239, the mechanism of training with 9 conditions is not clear enough. As different conditions have different levels of sparse information of input images, why they have equal training iterations? And continuous shifting between different conditions may make the training hard.

3. the motivation why the new conditions are not trained as the Base ControlNet by a shifting mechanism is not clear enough.

4. Most results are from "Base CN + CtrlLoRA'', and results from "Community Model + CtrlLoRA" in Figure 11a are rare, not enough to convince that CtrlLoRA is effective when transferring to other community models.

5. pretrained-VAE seems to be only an interesting trick.

6. putting all the prompts in the appendix makes reading inconvenient.

**Questions:**

1. The results in Figure11b demonstrate that the different conditions are effectively disentangled, with a direct summation module according to Figure 3c. Could you clarify why this module is effective, such as presenting the results of two elements both separately and after sum-up.

2. A detail, why not presenting all 9 base-condition results comparison to UniControl in Table 2?

---

> ### Author Response · Authors · 2024-11-20
> **Response to Reviewer GM4E (Part 1/2)**
>
> We sincerely thank Reviewer GM4E for carefully reading our paper and giving a valuable review.
>
> &nbsp;
>
> > **W1:** In line 70, the authors state that ControlNet with Canny edges requires 3 million images over 600 GPU hours for one condition. In contrast, line 244 indicates that Base ControlNet necessitates millions of images for 6000 GPU hours for 9 conditions. Although it is not fair enough, but it implies that the proposed method does not significantly reduce the computational burden.
>
> We would like to clarify that our work aims at reducing the computational burden for any **new** conditions, but not for the base conditions (Base ControlNet). Given our pre-trained Base ControlNet, the community users just need to collect about 1000 data pairs of a customized condition, and then fine-tune the Base ControlNet with less than one hour on a single GPU. That is to say, we make efforts to train and release a Base ControlNet while letting the ordinary user create their customized ControlNet at a significantly low cost.
>
> ---
>
> &nbsp;
>
> > **W2:** In line 239, the mechanism of training with 9 conditions is not clear enough. As different conditions have different levels of sparse information of input images, why they have equal training iterations? And continuous shifting between different conditions may make the training hard.
>
> We agree that setting different weights (iterations) to different conditions may further enhance the model performance. However, ablating the weight choices is tough since training the Base ControlNet for once consumes a large amount of resources, while our devices cannot support such an amount of ablation. Therefore, without any prior knowledge of information density of different conditions, we just choose an equal weight for them and find it works well enough. Nonetheless, this is a very interesting perspective, and we will keep investigating this problem in future works.
>
> ---
>
> &nbsp;
>
> > **W3:** the motivation why the new conditions are not trained as the Base ControlNet by a shifting mechanism is not clear enough.
>
> When learning new conditions, we freeze the parameters of the Base ControlNet and only train the newly added LoRA layers. In this case, the learning of each condition is independent and only affects its own corresponding LoRA. Thus, there is no need to train the new conditions by the shifting mechanism.
>
> ---
>
> &nbsp;
>
> > **W4:** Most results are from "Base CN + CtrlLoRA'', and results from "Community Model + CtrlLoRA" in Figure 11a are rare, not enough to convince that CtrlLoRA is effective when transferring to other community models.
>
> In Figure 16 in the appendix, our CtrLoRA is applied to 7 community models of various styles for style transfer, demonstrating its adaptability to other community models.
>
> ---
>
> &nbsp;
>
> > **W5:** pretrained-VAE seems to be only an interesting trick.
>
> Using pretrained VAE is vital to alleviate the sudden convergence phenomenon and achieve fast convergence. We have made extensive analyses on the use of pretrained VAE as the condition embedding network, in Section 3.4, lines 412-415 (Table 4, Figure 7), and Appendix Section A (Figure 12). For example, as shown in Figure 7, without pretrained VAE, a ControlNet needs more than 40,000 steps to converge, while applying pretrained VAE shortens the convergence to 4,000 steps. This choice is built upon our deep understanding of the convergence problem of ControlNet, and we believe our analysis is useful and can benefit the ControlNet community.
>
> ---
>
> &nbsp;
>
> > **W6:** putting all the prompts in the appendix makes reading inconvenient.
>
> Since our method mainly focuses on image-to-image generation, the text prompts are of less importance. Therefore, we put the prompts in the appendix in order to better organize the figures and keep the presentation cleaner.

---

> ### Author Response · Authors · 2024-11-20
> **Response to Reviewer GM4E (Part 2/2)**
>
> > **Q1:** The results in Figure11b demonstrate that the different conditions are effectively disentangled, with a direct summation module according to Figure 3c. Could you clarify why this module is effective, such as presenting the results of two elements both separately and after sum-up.
>
> As explained in the LoRA paper [1], the LoRA layers work by amplifying existing knowledge in the base network. In our scenario, we have a speculation that the LoRAs trained on different conditions amplify different knowledge in the Base ControlNet by orthogonal semantic directions, therefore direct summation is natural for the model to handle both conditions simultaneously. It's a good suggestion to present the results of two elements both separately and after sum-up. *Since we cannot include figures in the comment, we will present these results in the revised paper*.
>
> ---
>
> &nbsp;
>
> > **Q2:** A detail, why not presenting all 9 base-condition results comparison to UniControl in Table 2?
>
> We didn't include all results only for typesetting consideration. The result for "Bbox" condition is presented below, which does not change our conclusion in line 336 of our paper: "*for base conditions, our base ControlNet performs on par with the state-of-the-art UniControl, demonstrating its robust fundamental capabilities*". We will include it in the revised paper.
>
> |                | Bbox              |
> | -------------- | ----------------- |
> |                | LPIPS↓ / FID↓     |
> | UniControl     | **0.292** / 26.65 |
> | CtrLoRA (ours) | 0.315 / **23.95** |
>
> ---
>
> &nbsp;
>
> [1] Hu, Edward J., et al. "LoRA: Low-Rank Adaptation of Large Language Models." International Conference on Learning Representations.

---

> > ### Comment · Reviewer_GM4E · 2024-11-23
> > **Further concerns about motivation**
> >
> > Thanks for your responses.
> >
> > Though I haven't seen a revised paper to clarify my Q1 yet, I believe this paper is good from an engineering perspective.
> >
> > However, I am curious about the practical value of your proposed CtrtLoRA. Given the wide availability of well-developed LoRA weights, such as those shared by the Civitar community, and the styles or modalities that typically require fine-tuning are less and less. Since people could access rich and enough LoRAs, they could simply apply them in their SD.
> >
> > Since the primary advantage of your approach seems to lie in speed rather than performance, and your new LoRA styles are not so customized enough, what is the motivation that we require this fast training technique?
> >
> > What's more, according to your paper and responses, some parts of the design are still not clear and convincing. I will change my score if concerns addressed.

---

> > > ### Author Response · Authors · 2024-11-23
> > > **To address the misunderstanding**
> > >
> > > Thank you for your response.
> > >
> > > &nbsp;
> > >
> > > > Though I haven't seen a revised paper to clarify my Q1 yet, I believe this paper is good from an engineering perspective.
> > >
> > > Some experimental results intended for the rebuttal remain incomplete; the revised version is scheduled to be uploaded in 24 hours.
> > >
> > > ---
> > >
> > > &nbsp;
> > >
> > >
> > > > However, I am curious about the practical value of your proposed CtrtLoRA. Given the wide availability of well-developed LoRA weights, such as those shared by the Civitar community, and the styles or modalities that typically require fine-tuning are less and less. Since people could access rich and enough LoRAs, they could simply apply them in their SD.
> > >
> > > > Since the primary advantage of your approach seems to lie in speed rather than performance, and your new LoRA styles are not so customized enough, what is the motivation that we require this fast training technique?
> > >
> > > Thank you for this comment. However, this is a complete misunderstanding of our paper.
> > >
> > > + Almost all the LoRAs for Stable Diffusion (SD) you can find on the internet (including Civitai) are trained for **Stylized Outputs**. For example, a LoRA can be trained to make the SD output pixel style, cartoon style, or pencil style images. **In other words, the LoRAs you are talking about are designed to change the "Output Domain" of SD**. As you mentioned, we can find a lot of this kind of LoRAs in the community.
> > > + However, the LoRAs in our paper is totally different from what you are talking about. The LoRAs in this paper are designed for adapting the Base ControlNet to various **Controlling Inputs**.  For example, a LoRA can be trained to make the Base ControlNet accept lineart images or depth images as input. **In other words, the LoRAs in our paper are designed to change the "Input Domain" of ControlNet.** This kind of LoRAs is not only rare in the community but also not well explored in the research area.
> > >
> > > We sincerely hope you can think carefully about the above difference and kindly read our paper again. Besides, we've clarified our motivation and contribution in the ["General Response"](https://openreview.net/forum?id=3Gga05Jdmj&noteId=v3zsgz6v6e) at the top of this page. We wish we could bring you a correct understanding of our work, and sincerely hope that your rating will not be based on this complete misunderstanding. Thank you very much again.

---

> > > > ### Comment · Reviewer_GM4E · 2024-11-28
> > > > **reply to authors GM4E**
> > > >
> > > > Thanks for your clarification and additional experiments, I once got confused by the community LoRAs. Overall I think this paper is good from an engineering perspective but my major concern is still the lack of theoretical validation, after reading other comments and rebuttals, I think "marginally above threshold" is suitable.

---

> ### Author Response · Authors · 2024-11-24
> **Paper Revision for Q1**
>
> We have uploaded a revised paper with details in ["Paper Updates"](https://openreview.net/forum?id=3Gga05Jdmj&noteId=isfjp1umdb) at the top of this page, including the response for **Q1** in Appendix Figure 17.
>
> > **Q1**: The results in Figure11b demonstrate that the different conditions are effectively disentangled, with a direct summation module according to Figure 3c. Could you clarify why this module is effective, such as presenting the results of two elements both separately and after sum-up.

---

### Official Review · Reviewer_UthF · 2024-11-03

**Soundness:** 3
**Presentation:** 3
**Contribution:** 3
**Rating:** 6
**Confidence:** 3

**Summary:**

The paper proposes CtrLoRA for better controllability of the conditional image generation. This framework trains a Base ControlNet for the general image-to-image generation and then uses the LoRA fine-tuning for specific user instructions. Experiments show the effectiveness of the proposed method.

**Strengths:**

- The paper is well-organized and easy to follow.
- The authors conduct sufficient ablation studies to evaluate the proposed modules.
- The experiments demonstrate the training efficiency of the proposed method and its capability to unify various visual conditions for generation.

**Weaknesses:**

- The authors train a base ControlNet for the subsequent LoRA fine-tuning. However, why not directly fine-tune a pre-trained ControlNet or Uni-ControlNet?

- Lack of comparison to: ControlNet++[1].

- The paper does not explore whether this method can be generalized to other diffusion models such as SDXL and Pixart.

[1] Li M, Yang T, Kuang H, et al. ControlNet++: Improving Conditional Controls with Efficient Consistency Feedback[C]//European Conference on Computer Vision. Springer, Cham, 2025: 129-147.

**Questions:**

Please refer to the weakness.

---

> ### Author Response · Authors · 2024-11-20
> **Response to Reviewer UthF**
>
> We sincerely thank Reviewer UthF for the precise comments and positive feedback.
>
> &nbsp;
>
> > **W1:** The authors train a base ControlNet for the subsequent LoRA fine-tuning. However, why not directly fine-tune a pre-trained ControlNet or Uni-ControlNet?
>
> While directly fine-tuning a pre-trained ControlNet or Uni-ControlNet is feasible, they do not perform as well as fine-tuning our Base ControlNet. As discussed in line 173 of our paper, a pre-trained ControlNet is extensively trained to fit a particular condition, and therefore not general enough to efficiently adapt to different conditions. For Uni-ControlNet [2] and UniControl [3], as discussed in line 161 of our paper, although they are trained on multiple conditions, their delicate design makes them not straightforward to be quickly extended to new conditions. On the contrary, our fine-tuning stage keeps consistent with the pre-training strategy of our Base ControlNet, and therefore the adaptation to new conditions is natural and efficient.
>
> Below we add the comparison to directly fine-tune a pre-trained ControlNet and UniControl on 1000 images. As can be seen, our CtrLoRA significantly outperforms these methods when adapting to new conditions, demonstrating the effectiveness of our Base ControlNet and the potential of our idea to learn the general knowledge of I2I generation.
>
> |                           | Lineart                     | Densepose                       | Inpainting                                | Dehazing                                  |
> | ------------------------- | --------------------------- | ---------------------------     | ----------------------------------------- | ----------------------------------------- |
> |                           | LPIPS↓ / FID↓               | LPIPS↓ / FID↓                   | LPIPS↓ / FID↓                             | LPIPS↓ / FID↓                             |
> | ControlNet (canny) + LoRA | 0.356 / $\underline{16.74}$ | 0.198 / $\underline{36.14}$               | 0.602 / 17.63                             | 0.618 / 51.55                             |
> | UniControl + LoRA         | $\underline{0.316}$ / 17.05 | $\underline{0.164}$ / 41.20     | $\underline{0.558}$ / $\underline{15.84}$ | $\underline{0.508}$ / $\underline{37.83}$ |
> | CtrLoRA (ours)            | **0.305** / **16.12**       | **0.159** / **35.18** | **0.326** / **9.972**                     | **0.255** / **15.44**                     |
>
> ---
> &nbsp;
>
> > **W2:** Lack of comparison to: ControlNet++ [1].
>
> We agree that including a comparison to ControlNet++ will make the analysis more comprehensive, and the corresponding results are presented below. It is reasonable that our method lags behind ControlNet++ because the latter is explicitly optimized by the metric functions. But note that this technique is orthogonal to our method and less relevant to our focus.
>
> |                | Canny             | Depth             | Segmentation          | Lineart           |
> | -------------- | ----------------- | ----------------- | --------------------- | ----------------- |
> |                | LPIPS↓ / FID↓     | LPIPS↓ / FID↓     | LPIPS↓ / FID↓         | LPIPS↓ / FID↓     |
> | ControlNet++   | **0.354** / 21.99 | **0.205** / 20.12 | **0.438** / **19.99** | **0.172** / 35.24 |
> | CtrLoRA (ours) | 0.388 / **16.65** | 0.222 / **19.34** | 0.465 / 21.13         | 0.247 / **13.47** |
>
> ---
> &nbsp;
>
> > **W3:** The paper does not explore whether this method can be generalized to other diffusion models such as SDXL and Pixart.
>
> We agree that it is important and useful to apply our method to more powerful backbones such as SDXL and Pixart. However, the development and extensive analysis of our Base ControlNet on SD 1.5 have exhausted our available devices (only 8~12 RTX 4090 GPUs with 24GB VRAM); we lack sufficient resources for larger backbones.
>
> Nonetheless, the whole design philosophy of our CtrLoRA, especially the training strategy, is not restricted to the current SD 1.5 backbone. Therefore, we believe our method and its advantages have the potential to be generalized to larger backbones (just like ControlNet, originally built upon SD 1.5, it is well generalized to various backbones). Of course, we would like to develop our method upon more powerful backbones for future works when we have more devices.
>
> ---
> &nbsp;
>
> [1] Li, Ming, et al. "ControlNet++: Improving Conditional Controls with Efficient Consistency Feedback." European Conference on Computer Vision. Springer, Cham, 2025.
>
> [2] Zhao, Shihao, et al. "Uni-controlnet: All-in-one control to text-to-image diffusion models." Advances in Neural Information Processing Systems 36 (2024).
>
> [3] Qin, Can, et al. "UniControl: A Unified Diffusion Model for Controllable Visual Generation In the Wild." Advances in Neural Information Processing Systems 36 (2024).

---

> > ### Comment · Reviewer_UthF · 2024-11-23
> >
> > Thank you for your response. The authors have addressed most of the concerns I thought important for the paper's completeness. Consequently, I will maintain my score of "marginally above the acceptance threshold".

---

### Author Response · Authors · 2024-11-20
**General Response**

Dear Reviewers and Area Chair,

We sincerely appreciate your effort and valuable review. For a better understanding of our paper, we wish to clarify our motivation and contribution.


### **1. Our goal**

Although ControlNet is powerful and popular, developing it for **new conditions** is still an extremely heavy burden for an **ordinary user**, considering its huge consumption of data, GPUs, training time, and model sizes. To this end, our **goal is to provide an I2I foundation model and corresponding solution that allows ordinary users to create their customized ControlNets at an affordable cost** (similar to the role that Stable Diffusions plays in the T2I generation community).


### **2. Existing methods related to the goal**

Three categories of existing methods are most related to our goal:

+ Methods such as UniControl [1], Uni-ControlNet [2], controlnet-union [3] unify multiple conditions into one single model, decreasing the number of models. However, they lack a straightforward method that extends the unified model to new conditions with limited data and GPUs. Although we could naively use LoRA to fine-tune these models, they perform worse than our method due to the inconsistency between pre-training and fine-tuning.

+ Methods such as T2I-Adapter [4] and SCEdit [5] improve the training efficiency and decrease the model sizes. However, the data and GPU resources required to train these models are still unaffordable for ordinary users.

+ Community model ControlLoRA [6] directly trains LoRA with controlling input on Stable Diffusion, which seems to be the most affordable method. However, without a powerful base model like our Base ControlNet, this method cannot obtain satisfactory performance.


### **3. How far we step towards the goal**

Existing related methods still fall far short of this goal, whereas our method makes a significant advancement towards the goal. Through extensive testing, our method enables users to create a customized ControlNet with limited data (≈1000), only 1 GPU, and within 1 hour. Besides, the LoRA size is small (≈37M params.), making it easy to distribute and deploy. As far as we know, this is **the most affordable solution for ordinary users to develop their own ControlNets with satisfactory results, which represents our contribution to practical applications**.


### **4. The main challenge of the goal**

To achieve our goal, we employ the LoRA technique for new conditions, which "seems to be straightforward". However, LoRA itself is not the challenge; **the real challenge is how to make a new LoRA perform well with limited data (1000 images in our paper), as it is difficult for an ordinary user to collect a large customized dataset**. This challenge is not straightforward to solve and existing methods cannot handle it well. To this end, we propose to train a Base ControlNet with a shifting condition scheme to capture the general knowledge of I2I generation. With our well trained Base ControlNet, 1000 data samples are sufficient to learn a LoRA for a new conditon with satisfactory results.


### **5. Why is our Base ControlNet better for learning new LoRAs**

Our Base ControlNet is trained with shifting base conditions, and these conditions themselves correspond to condition-specific LoRAs. In other words, the Base ControlNet is trained to be adapted for the LoRA form. Therefore, it's naturally suitable for learning new LoRAs.

---

[1] Qin, Can, et al. "UniControl: A Unified Diffusion Model for Controllable Visual Generation In the Wild." Advances in Neural Information Processing Systems 36 (2024).

[2] Zhao, Shihao, et al. "Uni-controlnet: All-in-one control to text-to-image diffusion models." Advances in Neural Information Processing Systems 36 (2024).

[3] xinsir, et al. "controlnet-union-sdxl-1.0." Hugging Face.

[4] Mou, Chong, et al. "T2i-adapter: Learning adapters to dig out more controllable ability for text-to-image diffusion models." Proceedings of the AAAI Conference on Artificial Intelligence. Vol. 38. No. 5. 2024.

[5] Jiang, Zeyinzi, et al. "Scedit: Efficient and controllable image diffusion generation via skip connection editing." Proceedings of the IEEE/CVF Conference on Computer Vision and Pattern Recognition. 2024.

[6] Wu, Hecong. "ControlLoRA: A Lightweight Neural Network To Control Stable Diffusion Spatial Information." GitHub.

---

### Author Response · Authors · 2024-11-24
**Paper Updates**

Dear Reviewers and Area Chair,

Thanks for your precious and careful review. We have uploaded a revised paper based on your feedback, with updates below:

**Major updates**
- **Related Work & Appendix Section B:** add discussions on T2I-Adapter, SCEdit, and ControlLoRA. [Reviewer **YjAj**]
- **Appendix Section C:** add more quantitative results, including a controllable generation benchmark [Reviewer **YjAj**, **97kn**], comparison with directly fine-tuning a pretrained ControlNet or UniControl [Reviewer **UthF**, **YjAj**, **97kn**], and the effect of the number of base conditions [Reviewer **YjAj**, **97kn**].
- **Appendix Figure 17:** show more visual results of multi-conditional generation, with both separate and sum-up results of two conditions. [Reviewer **GM4E**]

**Other updates**
- **Table 2 & Appendix Figure 14:** add the quantitative and visual results of Bounding Box condition. [Reviewer **GM4E**]
- **Table 3 & 4:** fix the LPIPS results of Lineart condition since the results in the first version is not accurate. This update does not affect any conclusions.

---

### Meta-Review · Area_Chair_LJ8i · 2024-12-17

**Metareview:**

This paper proposes proposes a novel approach to augment text-to-image diffusion models with spatial control for multiple tasks. A base controlNet is learned, which is adapted to different tasks using low-rank adapters for each task, facilitating adapting to new tasks with additional LoRA components using relatively few examples.
Strengths of the paper mentioned in the reviews include: paper organization, good ablations, efficient aggregation of several conditioning forms in single model + LoRA.
Weaknesses: why need the base controlnet, missing comparison to several baselines, no exploration of other diffusion backbones (SDXL, PixArt-alpha), lack of clarity in places.

**Additional Comments On Reviewer Discussion:**

In response to the reviews the authors submitted a rebuttal and a revised manuscript. The rebuttal addressed most concerns raised by the reviewers, as acknowledges by all four reviewers. The reviewers unanimously recommend accepting the paper, and the AC follows their recommendation.

---

### Decision · Program_Chairs · 2025-01-22

Accept (Poster)